# THE CURSE OF MULTI-MODALITIES: EVALUATING HALLUCINATIONS OF LARGE MULTIMODAL MODELS ACROSS LANGUAGE, VISUAL, AND AUDIO

## ABSTRACT

Recent advancements in large multimodal models (LMMs) have significantly enhanced performance across diverse tasks, with ongoing efforts to further integrate additional modalities such as video and audio. However, most existing LMMs remain vulnerable to hallucinations, the discrepancy between the factual multimodal input and the generated textual output, which has limited their applicability in various real-world scenarios. This paper presents the first systematic investigation of hallucinations in LMMs involving the three most common modalities: language, visual, and audio. Our study reveals two key contributors to hallucinations: overreliance on unimodal priors and spurious inter-modality correlations. To address these challenges, we introduce the benchmark *The Curse of Multi-Modalities* (**CMM**), which comprehensively evaluates hallucinations in LMMs, providing a detailed analysis of their underlying issues. Our findings highlight key vulnerabilities, including imbalances in modality integration and biases from training data, underscoring the need for balanced cross-modal learning and enhanced hallucination mitigation strategies. Based on our observations and findings, we suggest potential research directions that could enhance the reliability of LMMs. We will make our code and data publicly available.

## 1 INTRODUCTION

Large Multi-modal Models (LMMs) have rapidly advanced, driving significant improvements across a wide range of tasks by effectively integrating and processing diverse data modalities. These models (Li et al., 2024a; Zhang et al., 2024; Hong et al., 2024; Yao et al., 2024; Wang et al., 2024a; Achiam et al., 2023; Team et al., 2023; Ormazabal et al., 2024), leveraging multimodal inputs such as image and text, have achieved notable performance gains, particularly in generating contextually accurate textual outputs. As the field evolves, there is a growing trend toward incorporating additional modalities, such as audio and video (Xu et al., 2024; Chen et al., 2024a; Wang et al., 2024c; Zhang et al., 2023; Cheng et al., 2024; Li et al., 2024b; Kong et al., 2024; Tang et al., 2023; Ghosh et al., 2024; Chu et al., 2024), to enhance LMMs' ability to understand and interact with complex real-world environments. However, despite these advancements, LMMs are prone to a critical issue known as hallucination, where the generated outputs do not accurately reflect the multimodal inputs (Liu et al., 2023; Wang et al., 2023a; 2024d; Nishimura et al., 2024). This issue can severely undermine the reliability and applicability of LMMs in real-world scenarios, particularly in tasks requiring precise and factual content generation.

Hallucination, particularly object hallucination, has been a key focus in LMMs that handle image and text inputs. Object hallucination occurs when LMMs generate semantically coherent but factually unaligned contents with the actual objects present in the input images. Various benchmarks (Li et al., 2023; Sun et al., 2023b; Lovenia et al., 2023; Wang et al., 2023a) and mitigation techniques have been proposed to address this issue by refining training processes (Liu et al., 2023), implementing post-hoc correction (Leng et al., 2024; Zhou et al., 2023), etc. However, accommodating additional modalities like audio and video exacerbates alignment and fusion difficulties (Lahat et al., 2015; Dimitri, 2022; Tong et al., 2024; Liang et al., 2024), which may lead to increased hallucinations.

This study systematically examines how LMMs produce hallucinations while integrating language, visual, and audio inputs, revealing the prevalence and causes of hallucinations under such multimodal scenarios. Two key contributors are identified: (1) Overreliance on unimodal priors: Models over-rely on data from a single modality, neglecting others. This results in outputs that do not accurately reflect the full range of input data, as models default to familiar patterns within one modality despite contradictory signals from others. (2) Spurious inter-modality correlations: Models learn erroneous cross-modal associations based on patterns that appear statistically significant but lack meaningful or causal connections, leading to plausible but counterfactual outputs. We introduce *The Curse of Multi-Modalities* (**CMM**), a comprehensive benchmark for assessing hallucinations in LMMs, covering a wide range of scenarios across visual, audio, and their joint contexts. CMM converts hallucination evaluation into a binary classification task through object-level and event-level probing. It comprises $1,200$ video/audio/video-audio samples across various multimodal contexts, ensuring balanced evaluation with $2,400$ probing questions evenly split between queries for existent and non-existent objects/events. LMMs are prompted with straightforward yes-or-no questions regarding the presence of specific objects or events in the input modalities.

CMM is the first benchmark to systematically investigate LMMs' hallucinations in such comprehensive multimodal settings. Unlike prior benchmarks that broadly assess hallucination performance, CMM segments hallucinations into nuanced subcategories under two key contributors: spurious inter-modality correlations (e.g., Visual-Language, Audio-Language, Visual-Audio-Language) and unimodal overreliance (e.g., Language Domianance, Visual Dominance, Audio Dominance), enabling precise diagnosis of LMM vulnerabilities and shedding light on possible improvements. By introducing diagnostic metrics including perception accuracy (PA) and hallucination resistance (HR), CMM offers a comprehensive framework for gauging both perception capabilities and hallucination severity in LMMs. In summary, the contributions of this work are threefold:

- We conduct the first systematic investigation of hallucinations in LMMs across language, visual, and audio modalities, identifying their key contributors including unimodal prior overreliance and spurious inter-modality correlations.

- We introduce a novel and comprehensive benchmark, *The Curse of Multi-Modalities* (**CMM**), which evaluates hallucinations using object-level and event-level probing within a binary classification framework. CMM defines hallucinations with nuanced subcategories and granularities, enabling comprehensive diagnosis of LMM vulnerabilities across various modalities.

- We evaluate a diverse set of state-of-the-art LMMs across visual, audio, and joint contexts, revealing critical insights in model limitations and fundamental challenges in multimodal learning. Our thorough analysis and discussion pinpoint future directions for mitigating hallucinations and enhancing LMM reliability, providing a viable roadmap for improvements.

## 2 ANALYZING HALLUCINATIONS ACROSS LANGUAGE, VISUAL, AND AUDIO

This section systematically investigates the underlying causes of hallucinations in Large Multimodal Models (LMMs). It includes qualitative demonstrations and comprehensive statistical analysis from two key perspectives: *Overreliance on Unimodal Priors* and *Spurious Inter-modality Correlations*. Our analysis provides empirical evidence and quantifies the extent to which these factors influence LMMs' reliability.

**Notations.** Consider an LMM parametrized by $\theta$ that processes inputs from three modalities: language $x$, visual $v$, and audio $a$. The model generates textual output $y$ autoregressively, where each token $y_t$ is conditioned on all three modalities and the previously generated tokens $y_{<t}$:

$$y_t \sim p_\theta(y_t \mid v, a, x, y_{<t}),$$

where $y_t$ represents the token at time step $t$, and $y_{<t}$ denotes the sequence of tokens generated up to time step $t - 1$.

### 2.1 OVERRELIANCE ON UNIMODAL PRIORS

Overreliance on unimodal priors is a key factor contributing to hallucinations in LMMs. This issue arises when the model over-relies on the knowledge learned from one modality during training,

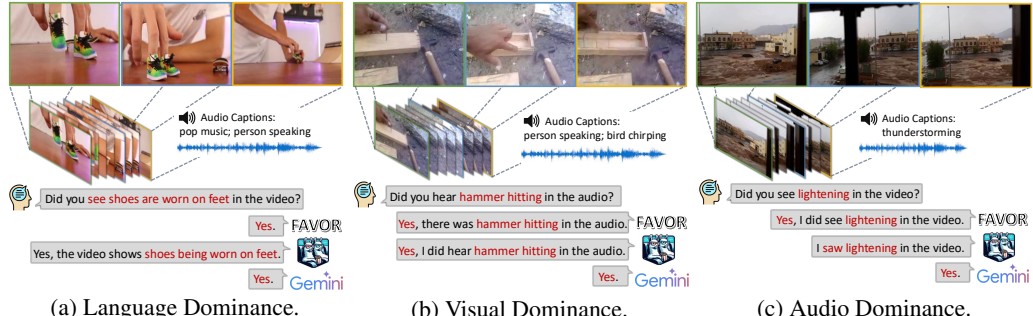

Figure 1: Demonstrations of overreliance on unimodal priors.

rather than integrating knowledge of all available modalities. In such cases, the model defaults to strong unimodal priors learned during training, leading to outputs that follow familiar unimodal patterns even when those patterns are not supported by the multimodal input. Following the general issue of overreliance on unimodal priors, we categorize this into three distinct types: **Language Dominance**, **Visual Dominance**, and **Audio Dominance**. Each form of dominance presents unique challenges for LMM performance and contributes to hallucinations in different ways.

**Language Dominance**, also know as language biases (Rohrbach et al., 2018; Leng et al., 2024; Guan et al., 2024; Wang et al., 2024b), arises when a model excessively depends on pre-trained large language models (LLMs), generating responses that adhere to linguistic patterns or prior knowledge from large language corpora, even when visual or audio inputs provide contradictory information. This issue is particularly prevalent in LMMs that integrate LLMs as their decoder base. These LLMs Chiang et al. (2023); Jiang et al. (2023); Yang et al. (2024), due to their proficiency in capturing linguistic structures and semantic relationships, often dominate the decision-making process, overshadowing contributions from visual or audio modalities. As illustrated in Fig. 1a, a video depicts finger skateboarding with shoes on fingers. When asked by the language-biased question "Did you see shoes worn on feet?"—reflecting commonsense event that follows linguistic priors—LMMs respond "yes," contradicting the actual content and inducing hallucination. This demonstrates LMMs' tendency to rely on language priors over factual multimodal inputs.

**Visual Dominance** occurs when a model over-relies on visual information, underutilizing or disregarding linguistic and auditory cues. In such cases, the model-generated outputs are heavily influenced by visual context, often neglecting important information from the other modalities. As illustrated in Fig. 1b, a video depicts a person planning a woodworking project with a hammer in sight, while the audio track contains only the person speaking and bird chirping. Despite this, advanced LMMs may over-rely on the visual presence of the "hammer" and incorrectly infer a "hammer hitting" sound, ignoring the actual audio content where no such sound is present.

**Audio Dominance** arises when a model excessively relies on auditory input, disregarding visual or linguistic information. As illustrated in Fig. 1c, a video captures a person recording a village view through a window, showing dark clouds. The audio track contains evident thunderstorm sounds, but no lightning is visible. Despite this, LMMs may over-rely on the audio cues, hallucinating that lightning is visible in the scene, thereby disregarding the actual visual content.

To validate our observations on unimodal overreliance, we conduct case studies on each example in Fig. 2, hypothesizing that altering information from a dominant modality would significantly affect the model's responses if hallucinations are primarily due to overreliance on that modality.

In the visual dominance scenario, we progressively blur the video to reduce visual content and tracked the probabilities of the LMM responding with a hallucinatory "yes" ($p_\theta$("yes" $\mid v', a, x$)) or a correct "no" ($p_\theta$("no" $\mid v', a, x$)) across different blur levels. As shown in Fig. 2b, increasing the blur led to a significant decline in hallucinatory "yes" responses and a rise in correct "no" responses. This indicates that reducing visual information compels the model to rely more on auditory cues, thereby decreasing visual-induced hallucinations. In the audio dominance case (Fig. 2c), we add noise to the audio track to degrade its quality. As noise levels increased, the probability of hallucinatory "yes" responses decreased, while correct "no" responses became more frequent ($p_\theta$("yes"/"no" $\mid v, a', x$)). This demonstrates that diminishing auditory information shifts the model's reliance to visual cues, mitigating hallucinations caused by overreliance on auditory inputs. For the language dominance scenario, we blur the video containing critical visual information

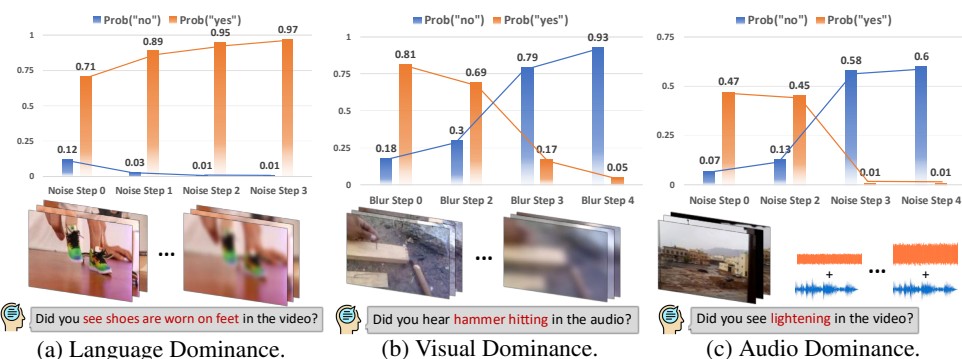

Figure 2: Validation experiments on overreliance on unimodal priors.

needed to accurately answer an adversarial question. As the visual content was increasingly obscured, the model's reliance on language priors intensifies, leading to more hallucinatory "yes" responses and fewer correct "no" responses (Fig. 2a). This suggests that in the absence of visual details, the model defaults to language-based patterns, exacerbating hallucinations.

In summary, these case studies confirm that unimodal overreliance significantly contributes to hallucinations in LMMs. Reducing information from the dominant modality forces the model to integrate cues from other modalities more effectively, thereby decreasing the likelihood of hallucinations. This validates the challenges posed by uni-modality overreliance in multimodal integration.

## 2.2 SPURIOUS INTER-MODALITY CORRELATIONS

Spurious inter-modality correlations are a major contributor to hallucinations in LMMs, especially when integrating multiple modalities. Learned during pretraining on large-scale multimodal datasets (e.g., image-caption, video-caption, and audio-caption data (Lin et al., 2014; Kim et al., 2019; Bain et al., 2021; Schuhmann et al., 2022; Wang et al., 2023b; Sun et al., 2024)), these correlations involve misleading associations between modalities that appear statistically significant but lack meaningful or causal connections. Two common sources of spurious correlations are: (1) Global occurrence frequency: The high overall occurrence of specific objects or events in the dataset leads LMMs to hallucinate these elements even when they are absent in the input. (2) Co-occurrence frequency: Frequent co-occurrence of objects or events during training causes the model to incorrectly predict the presence of one of them when only the other is present. While spurious object-level correlations between language and visual inputs have been extensively studied (Rohrbach et al., 2018; Li et al., 2023; Zhou et al., 2023), integrating additional modalities like audio introduces new complexities, resulting in increasingly intricate spurious correlations. We categorize them into three subtypes:

- **Visual-Language (VL):** The model hallucinates visual objects or events based on pre-training patterns. For instance, if "phone" frequently co-occurs with "human" in captions, the model may hallucinate a phone upon recognizing a human, even when no phone is present.

- **Audio-Language (AL):** The model links absent sound events to textual descriptions due to over-represented patterns in pre-training data. For example, if "dog barking" frequently appears during pre-training, the model may hallucinate this audio event even when the dog in the current input simply whimpers [1].

- **Visual-Audio-Language (VAL):** Spurious correlations arise from frequent co-occurrence of visual objects and audio events in video-audio joint training. For example, if "bird chirping" in audio descriptions is often paired with "tree" in visual annotations, the model may hallucinate to see trees when only hearing birds, or vice versa.

To validate spurious inter-modality correlations, we curate 200 samples for each subtype, paired with probing questions that target non-existent objects or events based on learned co-occurrence patterns. For VL, video-only samples are paired with questions about non-existent objects that frequently

---

[1]It is worth noting that, due to the sparsity of video-caption and audio-caption data—where typically only a single event is described per caption—event-level spurious correlations driven by co-occurrences for Visual-Language and Audio-Language often form between specific objects and their associated action-subject pairs.

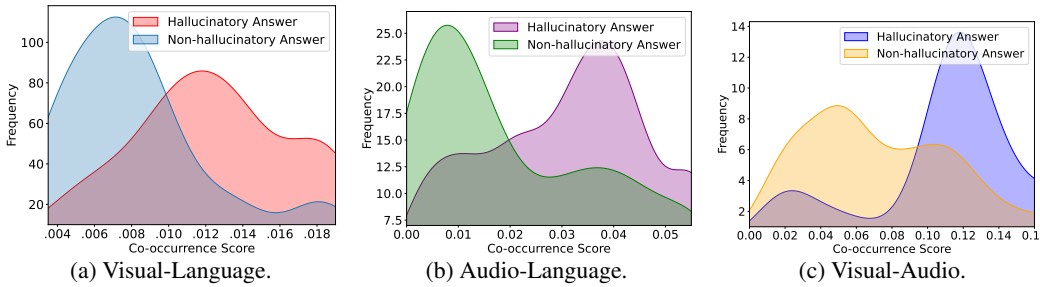

(a) Visual-Language.    (b) Audio-Language.    (c) Visual-Audio.

Figure 3: Validation experiments on spurious inter-modality correlations caused by co-occurrences.

co-occur. In AL, all queries are event-level, targeting absent audio events while a co-occurring action-object pair is present (e.g., querying "dog barking" when the dog only whimpers). For VAL, video-audio pairs are probed for non-existent visual objects or audio events based on frequently co-occurring pairs. We adopt the Co-occurrence Score (CoScore) from previous work (Biten et al., 2022; Zhou et al., 2023) to quantify co-occurrence frequency:

$$\text{CoScore}_s = \sum \frac{|S(o_{s,i}) \cap S(o_{s,j})|}{|S(o_{s,i})| + |S(o_{s,j})|},$$

where $S(o_{s,i})$ denotes the set of captions mentioning the $i$-th object or event within a sample $s$. Three open-source LMMs (FAVOR (Sun et al., 2023a), GroundingGPT (Li et al., 2024c), VideoLLaMA2-7B (Cheng et al., 2024)) are evaluated, with aggregated results shown in Fig. 3, plotting CoScore against the frequency of hallucinatory and non-hallucinatory answers. A consistent trend emerges: hallucinatory responses are associated with higher CoScores, indicating that higher co-occurrence frequencies increase the likelihood of hallucinations. This confirms the impact of spurious inter-modality correlations learned during pretraining[2].

## 3 CMM BENCHMARK: THE CURSE OF MULTI-MODALITIES

| Overreliance on Unimodal Priors | | | Spurious Inter-modality Correlations | | |
|---|---|---|---|---|---|
| Visual Dominance | Audio Dominance | Language Dominance | Visual-Language | Audio-Language | Visual-Audio-Language |

Table 1: Overall composition of CMM.

Inspired by the findings in previous section, we introduce *The Curse of Multi-modality* (**CMM**) benchmark, designed to systematically evaluate hallucinations in LMMs from two key contributors: Overreliance on Unimodal Priors and Spurious Inter-modality Correlations. As shown in Tab. 1, each type is further divided into specific sub-categories, enabling fine-grained assessment of how these factors influence LMMs' performance.

### 3.1 DATA COMPOSITION AND EVALUATION SETUP

For each subcategory, we manually collect 200 samples (video-only, audio-only, or video-audio pairs) to evaluate LMMs' handling of multimodal inputs. Each sample includes two modality-specific probing questions: one targeting a non-existent object or event (ground-truth answer "no") and one targeting an existent object or event (ground-truth answer "yes"):

*"Did you see [object/event] in the video?"*, for visual queries
*"Did you hear [event] in the audio?"*, for audio queries

This results in a total of $1,200$ samples and $2,400$ probing questions. We benchmark LMMs using two core metrics, namely, Perception Accuracy (PA) and Hallucination Resistance (HR):

$$\text{PA} = \frac{\#\text{correctly predicted "yes"}}{\#\text{ground-truth "yes"}} \qquad \text{HR} = \frac{\#\text{correctly predicted "no"}}{\#\text{ground-truth "no"}}$$

PA measures the model's ability to accurately perceive present objects or events, while HR assesses its resistance to hallucinations by correctly identifying the absence of non-existent objects or events. Higher scores in both metrics indicate better perception and robustness against hallucinations.

---

[2]Further experimental details for analysis are provided in the Appendix A.1.

## 3.2 DATA CONSTRUCTION

### 3.2.1 CONSTRUCTING QUERIES FOR OVERRELIANCE ON UNIMODAL PRIORS

To assess overreliance on a single modality (visual, audio, or language), we construct targeted probing queries that test the model's dependence on one modality while ignoring complementary signals.

**Visual Dominance** This subcategory tests whether LMMs hallucinate audio events based on visual input, where we construct queries asking about the existence of specific audio event. While queries with a "yes" answer are manually annotated, non-existent events are sourced from video-audio pairs in the AudioCaps dataset (Kim et al., 2019), where visual objects that do not correspond to any audio content are identified. These samples are manually verified to ensure accurate annotation, setting the ground truth answer as "no."

**Audio Dominance** We probe LMMs' tendency to infer incorrect visual content from audio cues. Queries ask about the presence of visual objects, with "yes" queries annotated manually. For "no" queries, we filter video-audio pairs from AudioCaps where audio-indicated objects have no visual representation, confirmed through manual review.

**Language Dominance** To explore how language priors contribute to hallucinations, we define sets of common-sense events (e.g., "fish swim in water") and object attributes (e.g., "yellow banana") to reflect typical linguistic biases. Videos are manually sourced from YouTube to depict anti-common-sense scenarios (e.g., "fish fly in the air," "black banana"). For existence-probing queries, we ask about the video's anti-common-sense object/event, annotating the ground truth as "yes." Conversely, for non-existence probing queries, we test for the common-sense version of the object/event, setting the ground truth as "no."

### 3.2.2 CONSTRUCTING QUERIES FOR SPURIOUS INTER-MODALITY CORRELATIONS

We evaluate hallucinations arising from *Spurious Inter-modality Correlations*, constructing object-level and event-level queries across visual, audio, and textual associations[3].

**Visual-Language** Hallucinations are assessed based on associations between visual content and textual descriptions. Object-level queries are derived from (i) global appearance frequencies and (ii) co-occurrence frequencies within the WebVid10M (Bain et al., 2021) video-caption dataset. Event-level queries, however, are constructed based on (i) global appearance patterns and (ii) [subject]-[action object] co-occurrence patterns. All probing samples are curated from WebVid10M.

**Audio-Language** Hallucinations derived from associations between audio and text are probed through event-level queries, given the temporal nature of audio. Queries are formed from (i) global appearance frequencies and (ii) subject-oriented co-occurrence patterns, based on data from the Auto-acd (Sun et al., 2024).

**Visual-Audio-Language** This subcategory explores hallucinations across visual and audio modalities. Queries probe non-existent audio events based on existent co-occurred visual objects and vice versa, with data sourced from AudioCaps (Kim et al., 2019), focusing on co-occurrence frequencies between visual objects and audio events.

## 4 EXPERIMENTS AND DISCUSSIONS

### 4.1 IMPLEMENTATION DETAILS

**Baselines** We evaluate a diverse set of LMMs on our benchmark, categorized into three groups based on their modality capabilities: models capable of processing both visual and audio inputs, visual-only models, and audio-only models.

- *Visual-Audio LMMs:* For models that process both visual and audio inputs, we include three proprietary models: Reka-core (Ormazabal et al., 2024), Gemini-1.5-flash (Team et al., 2023), and Gemini-1.5-pro (Team et al., 2023). In addition, we evaluate three open-source models: FAVOR-

---

[3]For more details on the construction process and data statistics, please refer to Appendix A.2 to A.4.

| Model | Spurious Inter-modality Correlation | | | | | | Uni-modality Overreliance | | | | | | Overall | |
|---|---|---|---|---|---|---|---|---|---|---|---|---|---|---|
| | VL | | AL | | VAL | | Visual Dom | | Audio Dom | | Lang Dom | | | |
| | pa ↑ | hr ↑ | pa ↑ | hr ↑ | pa ↑ | hr ↑ | pa ↑ | hr ↑ | pa ↑ | hr ↑ | pa ↑ | hr ↑ | pa ↑ | hr ↑ |
| Proprietary Models | | | | | | | | | | | | | | |
| Gemini-1.5-pro | 91.0 | 90.5 | 94.0 | 14.5 | 86.0 | 67.0 | 82.5 | 34.0 | 90.5 | 82.0 | 78.5 | 61.5 | 87.1 | 58.3 |
| Gemini-1.5-flash | 93.5 | 90.0 | 88.5 | 39.5 | 88.5 | 70.5 | 79.0 | 36.5 | 90.5 | 86.5 | 90.5 | 62.0 | 88.4 | 64.2 |
| Reka-core | 87.0 | 94.5 | 25.0 | 76.0 | 76.7 | 85.1 | 35.6 | 69.4 | 80.8 | 82.7 | 75.0 | 76.0 | 63.7 | 80.9 |
| Open-source Models | | | | | | | | | | | | | | |
| GroundingGPT | 95.5 | 36.5 | 100 | 0.0 | 97.5 | 18.0 | 99.5 | 1.0 | 98.5 | 23.5 | 88.5 | 7.0 | 96.6 | 14.3 |
| FAVOR | 91.0 | 55.0 | 94.5 | 45.0 | 94.5 | 69.0 | 89.0 | 21.5 | 92.0 | 43.5 | 92.0 | 18.5 | 92.2 | 42.1 |
| VideoLLaMA2 | 75.0 | 86.0 | 77.5 | 94.0 | 78.0 | 98.0 | 62.0 | 75.5 | 80.0 | 90.0 | 57.5 | 43.0 | 71.7 | 81.1 |

(a) Visual-Audio-Language LMMs results.

| Model | VL Correlations | | Lang Dominance | |
|---|---|---|---|---|
| | pa ↑ | hr ↑ | pa ↑ | hr ↑ |
| CogVLM2-Video | 99.50 | 44.00 | 98.00 | 5.00 |
| VideoChat2 | 97.00 | 66.00 | 88.00 | 34.50 |
| InternLM-XComposer 2.5 | 99.00 | 73.00 | 94.50 | 46.50 |
| PLLaVA | 89.50 | 93.00 | 75.00 | 52.00 |
| ShareGPT4Video | 87.50 | 85.50 | 79.50 | 58.00 |
| LLaVA-OneVision | 94.00 | 88.00 | 87.50 | 69.50 |
| GPT4o | 87.50 | 95.50 | 83.00 | 84.00 |

(b) Visual-Language LMMs results.

| Model | AL Correlations | |
|---|---|---|
| | pa ↑ | hr ↑ |
| Qwen2-Audio | 98.50 | 34.50 |
| Audio-Flamingo | 89.50 | 39.00 |
| GAMA-IT | 94.50 | 52.00 |
| SALMONN | 93.00 | 59.00 |

(c) Audio-Language LMMs results.

Table 2: Benchmarking results for LMMs across language, visual, and audio modalities.

13B (Sun et al., 2023a), GroundingGPT-7B (Li et al., 2024c), and VideoLLaMA2-7B (Cheng et al., 2024)[4].

- *Visual-Only LMMs:* For visual-only LMMs, we evaluate proprietary model GPT4o (OpenAI, 2024)[5]. We select several state-of-the-art open-source models for benchmarking, including VideoChat2-7B (Li et al., 2024b), ShareGPT4Video-8B (Chen et al., 2024a), PLLaVA-7B (Xu et al., 2024), CogVLM2-Video-19B (Hong et al., 2024), InternLM-XComposer2.5-7B (Zhang et al., 2024), and LLaVA-OneVision-7B (Li et al., 2024a).

- *Audio-Only LMMs:* For audio-only LMMs, since no such proprietary models are available, we focus on open-source models: Audio-Flamingo-1.3B (Kong et al., 2024), SALMONN-13B (Tang et al., 2023), GAMA-IT-7B (Ghosh et al., 2024), and Qwen2-Audio-7B (Chu et al., 2024).

**Evaluation Protocol** All models are evaluated using a sampling decoding strategy with a fixed temperature of 0.2 for consistency. We assess models based on Perception Accuracy (PA) and Hallucination Resistance (HR) metrics (see Sec. 3.1). Each model is prompted to "Answer with yes or no," and PA and HR are computed based on whether the response begins with "yes" or "no."

## 4.2 MAIN RESULTS

### 4.2.1 ANALYZING VISUAL-AUDIO LMMS

The results of LMMs that can process both visual and audio inputs are presented in Tab. 2a.

**Hallucination Vulnerability from Spurious Inter-Modality Correlations** Visual-Audio LMMs generally achieve PA scores over 80, demonstrating effective multimodal perception. Extensive efforts to mitigate Visual-Language (VL) spurious correlations have significantly reduced hallucinations, as proprietary models like Reka-core and Gemini-1.5 reach HR scores around 90. In contrast, open-source models like FAVOR and GroundingGPT continue to struggle with VL correlations.

However, the introduction of audio intensifies hallucinations across all models. Even Gemini-1.5-pro only attains a 14.5 HR score for Audio-Language (AL) correlations, highlighting the difficulty

---

[4]After extensive survey and reproduction efforts, we found these models to be accessible and reproducible.
[5]10 frames are uniformly sampled from each video and provided as input to GPT4o.

| Model | VL Correlations | | | | Language Dominance | | | |
|---|---|---|---|---|---|---|---|---|
| | object-level (pa/hr) | | event-level (pa/hr) | | object-level (pa/hr) | | event-level (pa/hr) | |
| Visual-Audio LMMs | | | | | | | | |
| Reka-core | 93.0 | 92.0 | 81.0 | 97.0 | 73.0 | 91.0 | 77.0 | 61.0 |
| Gemini-1.5-flash | 98.0 | 85.0 | 89.0 | 95.0 | 93.0 | 74.0 | 88.0 | 50.0 |
| Gemini-1.5-pro | 97.0 | 88.0 | 85.0 | 93.0 | 88.0 | 63.0 | 69.0 | 60.0 |
| FAVOR | 99.0 | 35.0 | 83.0 | 75.0 | 100 | 3.0 | 84.0 | 34.0 |
| GroundingGPT | 98.0 | 31.0 | 93.0 | 42.0 | 91.0 | 6.0 | 86.0 | 8.0 |
| VideoLLaMA2 | 76.0 | 85.0 | 74.0 | 87.0 | 69.0 | 37.0 | 46.0 | 49.0 |
| Visual-Only LMMs | | | | | | | | |
| VideoChat2 | 98.0 | 60.0 | 96.0 | 72.0 | 92.0 | 30.0 | 84.0 | 39.0 |
| ShareGPT4Video | 88.0 | 90.0 | 87.0 | 81.0 | 81.0 | 67.0 | 78.0 | 49.0 |
| PLLaVA | 91.0 | 92.0 | 88.0 | 94.0 | 76.0 | 70.0 | 74.0 | 34.0 |
| CogVLM2-Video | 99.0 | 48.0 | 100 | 40.0 | 99.0 | 5.0 | 97.0 | 5.0 |
| InternLM-XComposer 2.5 | 99.0 | 89.0 | 99.0 | 57.0 | 97.0 | 62.0 | 99.0 | 57.0 |
| LLaVA-OneVision | 98.0 | 89.0 | 90.0 | 87.0 | 92.0 | 82.0 | 83.0 | 57.0 |
| GPT4o | 97.0 | 94.0 | 78.0 | 97.0 | 90.0 | 91.0 | 76.0 | 77.0 |

Table 3: Visual-only benchmark subset results grouped by probing granularity.

in handling these correlations. Moreover, AL correlations cause more severe hallucinations than Visual-Audio-Language (VAL) correlations, likely due to the limited availability of visual-audio-language datasets compared to audio-language data. This imbalance may lead LMMs to form stronger spurious correlations between audio and language, leading to more frequent hallucinations when processing audio-only content.

**Hallucination Vulnerability from Uni-modality Overreliance** Models show solid perception capabilities across Uni-modality Overreliance subcategories, with high PA scores. However, a notable gap emerges when comparing PA and HR scores, highlighting hallucination challenges due to unimodal dependence. Visual Dominance, in particular, proves to be more problematic than Audio Dominance for most models. For instance, Gemini-1.5-flash achieves an HR of 86.5 in Audio Dominance but only 36.5 in Visual Dominance, suggesting that overreliance on visual input presents a more significant challenge. This can be attributed to the larger volume of visual training data and a visual-centric bias in video-audio joint datasets. Moreover, Language Dominance reveals the impact of LLM decoders, with steep declines in HR from PA scores, as seen in FAVOR dropping from 92.0 to 18.5. This indicates a strong reliance on language priors, suggesting a need to better balance multimodal integration.

**Response Tendencies of LMMs** Certain models display atypical response patterns. GroundingGPT tends to answer "yes" indiscriminately, leading to high PA but very low HR scores (e.g., 0 in AL correlations). This behavior suggests overconfidence or excessive human alignment during training, as also previously noted by other research (Li et al., 2023). In contrast, Reka-core and VideoLLaMA2 exhibit cautious tendencies, showing higher HR than PA in many cases and occasionally very low PA scores (e.g., Reka-core's 25.0 PA in AL correlations). This likely reflects safety alignment strategies to reject uncertain inputs with "no" responses. These contrasting response tendencies underscore the varied behavioral patterns in LMMs and highlight the need for more balanced training strategies that ensure accurate, context-dependent responses without overconfidence or excessive caution.

### 4.2.2 ANALYZING VISUAL-ONLY AND AUDIO-ONLY LMMS

Visual-only and audio-only LMMs show superior perception accuracy in their respective domains compared to Visual-Audio LMMs, as evidenced by higher PA scores in Tab.2b and Tab.2c. However, this advantage does not extend to mitigating hallucinations. Similar to Visual-Audio LMMs, single-modality models remain vulnerable to hallucinations caused by spurious inter-modality correlations. Despite previous efforts to address VL correlations, some models still exhibit poor HR scores, such as CogVLM2-Video, which scores 44. Furthermore, AL correlations pose even greater challenges, with audio-only LMMs scoring between 30 and 60 in HR, underscoring the insufficient mitigation of hallucinations in audio-text interactions, likely due to the limited attention this issue has received

| Model | VAL Correlations | | | |
|---|---|---|---|---|
| | object-level | | event-level | |
| | (pa/hr) | | (pa/hr) | |
| Reka-core | 96.6 | 86.7 | 57.1 | 83.5 |
| Gemini-1.5-flash | 94.0 | 92.0 | 83.0 | 49.0 |
| Gemini-1.5-pro | 92.0 | 90.0 | 80.0 | 44.0 |
| FAVOR | 94.0 | 85.0 | 95.0 | 53.0 |
| GroundingGPT | 96.0 | 35.0 | 99.0 | 1.0 |
| VideoLLaMA2 | 84.0 | 99.0 | 72.0 | 97.0 |

Table 4: Effects of probing modalities.

| Model Specs | | VL Cor | | Lang Dom | |
|---|---|---|---|---|---|
| Name | LLM Size | (pa/hr) | | (pa/hr) | |
| PLLaVA | Vicuna 7B | 89.5 | 93.0 | 75.0 | 52.0 |
| PLLaVA | Vicuna 13B | 86.5 | 96.5 | 75.5 | 65.0 |
| PLLaVA | Yi 34B | 91.0 | 94.5 | 75.5 | 74.0 |
| LLaVA-OneVision | Qwen2 0.5B | 96.5 | 91.5 | 81.0 | 55.0 |
| LLaVA-OneVision | Qwen2 7B | 94.0 | 88.0 | 87.5 | 69.5 |
| LLaVA-OneVision | Qwen2 72B | 84.5 | 93.5 | 89.5 | 75.5 |

Table 5: Effects of LLM decoder sizes in LMMs.

in prior research (Nishimura et al., 2024). Additionally, most Visual-only LMMs exhibit low HR scores for Language Dominance, hovering around 50. This indicates a strong reliance on language priors, leading to hallucinations when visual input conflicts with linguistic expectations. However, GPT4o demonstrates balanced performance, likely due to post-training safety alignment, which balances perception and cautious response, reducing hallucinations.

Overall, these findings not only emphasize the ongoing hallucination challenges in current LMMs but also reinforce our claim that Spurious Inter-modality Correlations and Unimodal Overreliance are two key factors driving hallucinations.

## 4.3 DISCUSSIONS

**Effects of Probing Granularities** Our benchmark includes both object-level and event-level probing questions across subcategories[6]. As shown in Tab. 3, most models show lower PA scores for event-level queries than object-level ones, highlighting the challenge posed by temporal complexity and the limited availability of event-oriented training data. For Visual-Language (VL) spurious correlations, event-level probing yields higher HR scores than object-level probing. This may be due to the scarcity of event-level annotations in visual-text pretraining data, while object-level annotations are more prevalent, fostering stronger spurious correlations. Conversely, within Language Dominance under Unimodal Overreliance, HR scores are lower for event-level queries. This pattern is likely due to the autoregressive nature of large language models, which increases reliance on language priors as the length of processed sequences grows, heightening the risk of hallucinations, especially when longer event-related common-sense knowledge is involved.

**Effects of Probing Modalities** The Visual-Audio-Language (VAL) subcategory examines spurious correlations arising from the co-occurrence of visual objects and audio events. It includes two probing types: (1) object-level queries about non-existent visual objects when frequently co-occurring audio events are present, and (2) event-level queries about non-existent audio events when frequently co-occurring visual objects are present. Despite both probing types originating from similar co-occurrence patterns, HR scores for event-level (audio) probing are significantly lower than those for object-level (visual) probing across all models (Tab. 3). This finding aligns with Sec. 4.2.1's analysis of Visual and Audio Dominance under Unimodal Overreliance, suggesting a bias towards visual data due to its abundance in training and the visual-centric nature of joint visual-audio pretraining. As a result, models tend to over-rely on visual cues, leading to more pronounced hallucinations when predicting non-existent audio events.

**Effects of LLM Sizes** We analyzed the impact of LLM decoder sizes on two LMMs, PLLaVA and LLaVA-OneVision[7]. As shown in Tab. 5, increasing the LLM size has minimal influence on HR scores for Visual-Language spurious correlations, supporting our claim that these correlations primarily arise from global appearance and co-occurrence patterns in training data. In contrast, larger LLM sizes consistently improve HR scores for Language Dominance. For example, LLaVA-OneVision's HR score increases from $55.0$ ($0.5B$ LLM) to $75.5$ ($34B$ LLM), suggesting that larger LLMs are more adept at managing complex or contradictory multimodal inputs. Smaller LLMs, however, are more susceptible to overfitting to linguistic priors, leading to higher hallucination rates when faced with content that deviates from expected patterns.

---

[6]Audio-related subcategories exclusively contain event-level queries due to their temporal nature.

[7]To the best of our knowledge, these are the only models available in multiple sizes.

**Future Directions** Our analysis identifies key vulnerabilities in current LMMs, representing only a subset of broader challenges. These include but are not limited to unbalanced cross-modal integration, often with visual dominance overshadowing audio or text cues; spurious inter-modality correlations arising from training biases; overreliance on linguistic priors from large-scale LLM pretraining; and divergent response tendencies—either overconfident approval or overly cautious rejection. To address these challenges, we propose several potential directions for reference:

- Balanced Multi-modal Training Data: Creating datasets with balanced modality representation and diverse temporal annotations to reduce visual biases and improve event-level understanding.

- Advanced Cross-modal Fusion: Implementing dynamic fusion strategies to adjust modality importance based on context can improve multimodal integration and reduce hallucination.

- Mitigating Linguistic Priors: Fine-tuning LMMs with contextually diverse prompts and incorporating visual/audio fact-checking mechanisms can decrease overreliance on language priors.

- Refined Safety Alignment: Establishing balanced response strategies to avoid overconfidence or excessive caution ensures accurate interpretation, even for ambiguous inputs.

## 5 RELATED WORKS

**Large Multimodal Models** Recent advances in large multimodal models (LMMs) have focused on leveraging large language models (LLMs) as decoder bases to process complex image-text interactions. Models like LLaVA (Liu et al., 2024) and Flamingo (Alayrac et al., 2022) utilize transformer architectures to enhance cross-modal understanding, enabling nuanced visual-text comprehension for tasks such as visual question answering and image-based dialogue. Beyond static image-text tasks, recent approaches have aimed to extend multimodal capabilities by incorporating additional modalities like video and audio (Cheng et al., 2024; Wang et al., 2024c; Chu et al., 2024; Tang et al., 2023), fostering richer context and enhancing the model's ability to handle a diverse range of multimodal scenarios.

**Hallucinations in LMMs** Hallucination, particularly object hallucination, has been extensively studied in LMMs that process image and text. This phenomenon arises when a model generates content inconsistent with the actual objects present in the input image. Various benchmarks have been developed to assess hallucination in vision-language tasks (Li et al., 2023; Wang et al., 2023a; Guan et al., 2024; Nie et al., 2024; Chen et al., 2024b; Ye-Bin et al., 2024; YWang et al., 2024), and several mitigation techniques have been proposed (An et al., 2024; Leng et al., 2024; Huang et al., 2024; Yu et al., 2024; Sun et al., 2023b). However, research on hallucinations in LMMs beyond image-text tasks is scarce, with limited investigation into hallucinations involving additional modalities like audio and video (Wang et al., 2024d; Nishimura et al., 2024). Motivated by this gap, our work introduces the *Curse of Multi-modality* (**CMM**) benchmark, the first to systematically evaluate hallucinations across language, visual, and audio inputs. CMM provides a comprehensive evaluation framework to explore how LMMs handle complex multimodal integration, offering insights into model vulnerabilities and guiding the development of more reliable multimodal systems.

## 6 CONCLUSIONS

To the best of our knowledge, this paper is the first to systematically investigate and verify the two key contributors to hallucinations in large multimodal models (LMMs) across language, visual, and audio modalities: overreliance on unimodal priors and spurious inter-modality correlations. We introduce the *Curse of Multi-modality* (**CMM**) benchmark, which features nuanced subcategories and granularities along with diagnostic metrics, enabling precise diagnosis of model limitations and guiding targeted improvements. By benchmarking various LMMs across diverse multimodal contexts, we identified key vulnerabilities in current models, such as unbalanced multimodal integration and biases arising from pretraining datasets. Our analyses provide fundamental insights into multimodal learning, highlighting the need for improved alignment across multimodal inputs and offering foundational guidance for developing more robust and reliable LMMs. We conclude by outlining potential future directions, hoping to inspire subsequent research in this area.

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

| Category | ‖ | Overreliance on Unimodal Priors | | |
|---|---|---|---|---|
| **Sub-category** | ‖ | Visual Dominance | Audio Dominance | Language Dominance |
| **Modality** | ‖ | Visual+Audio | Visual+Audio | Visual |
| **Granularities** | ‖ | event-level | object-level | object-, event-level |

Table 6: Overview of CMM subcategories under Overreliance on Unimodal Priors, presenting their involved modalities, and probing granularities.

| Category | ‖ | Spurious Inter-modality Correlations | | |
|---|---|---|---|---|
| **Sub-category** | ‖ | Visual-Language | Audio-Language | Visual-Audio-Language |
| **Modality** | ‖ | Visual | Audio | Visual+Audio |
| **Granularities** | ‖ | object-, event-level | event-level | object-, event-level |

Table 7: Overview of CMM subcategories under Spurious Inter-modality Correlations, presenting their involved modalities, and probing granularities.

## A  APPENDIX

### A.1  EXPERIMENTAL DETAILS FOR ANALYZING HALLUCINATIONS

#### A.1.1  QUALITATIVE DEMONSTRATIONS

For the demonstrations in Fig. 1, we use three advanced LMMs capable of processing both visual and audio inputs: Gemini-1.5-pro (Team et al., 2023), FAVOR-13B (Sun et al., 2023a), and VideoLLaMA2-7B (Cheng et al., 2024). The case studies presented in Fig. 2 analyze hallucination tendencies by computing $p_\theta$("yes"/"no" $\mid v, a', x$) and $p_\theta$("yes"/"no" $\mid v', a, x$), using VideoLLaMA2-7B as a representative model.

#### A.1.2  QUANTITATIVE VALIDATION

For the quantitative validation shown in Fig. 3, we curate 200 samples for each subcategory of hallucination.

**Visual-Language Experiments:** Each sample is a video-only raw file associated with a probing question targeting the existence of a non-existent object, while a frequently co-occurring object is present. The co-occurrence scores are computed from the WebVid-10M dataset (Bain et al., 2021), from which the video samples are also sourced. For instance, a video containing a bird is queried with "Did you see trees in the video?" since "bird" and "tree" frequently co-occur in the pretraining data, although no tree is visually present in the sample.

**Audio-Language Experiments:** Given the temporal nature of audio, all queries are event-level. Each audio-only raw file is associated with a question about a non-existent audio event, while the subject of a related event can be recognized. For example, a dog whimpering is queried with "Did you hear dog barking?" Co-occurrence scores are computed from the audio-text pretraining dataset Auto-acd (Sun et al., 2024), which also provides the audio samples.

**Visual-Audio Experiments:** The co-occurrence scores are derived from the video-audio dataset AudioCaps (Kim et al., 2019), containing video samples with corresponding audio tracks. Each sample is queried about a non-existent visual object with a frequently co-occurring audio event, or vice versa.

The above experiments are conducted on three open-source LMMs that support both visual and audio inputs: FAVOR-13B (Sun et al., 2023a), GroundingGPT-7B (Li et al., 2024c), and VideoLLaMA2-7B (Cheng et al., 2024). The frequencies displayed in Fig. 3 represent the aggregated results across all three models.

## A.2 FREQUENT PATTERNS IN PRETRAINING DATASETS

The following outlines the frequent global appearances and co-occurrence patterns derived from major pretraining datasets, which is used to construct our benchmark.

---

**Patterns in Pretraining Datasets**

**Visual-Language Correlations from WebVid-10M**

- *Object-level*
    - **Top appeared objects**: [beach, boat, car, city, flower, mountain, person, phone, tree, water]
    - **Top co-occurrences**: [beach-person, car-person, city-person, dog-person, food-person, laptop-person, mountain-person, phone-person, tree-person, water-person]
- *Event-level*
    - **Top appeared events**: [person drinks coffee, person drives car, person eats food, person holds glass, person reads book, person rides bike, person uses camera, person uses laptop, person uses phone, person uses tablet]
    - **Top co-occurred (subject)-(action object) pairs**: [person-drinks coffee, person-drives car, person-eats food, person-holds glass, person-reads book, person-rides bike, person-uses camera, person-uses laptop, person-uses phone, person-uses tablet]

**Audio-Language Correlations from Auto-acd**

- *Event-level (since audio is inherently temporal)*
    - **Top appeared events**: [bird chirps, car passes, car revs, crowd cheers, dog barks, guitar strums, person laughs, person sings, person speaks, water splashes]
    - **Top co-occurred (subject)-(action object) pairs**: [car-honks, car-passes, car-revs, dog-barks, dog-howls, dog-whimpers, person-cheers, person-laughs, person-sings, person-speaks]

**Visual-Audio-Language from AudioCaps**

- *Cross-modality (visual object)-(audio event) co-occurrences*
    - **Top co-occurrences**: [person-bird chirping, tree-bird chirping, tree-car passing, person-dog barking, car-person speaking, table-person speaking, tree-person walking, person-water splashing, dog-person speaking, person-car revving, water-person speaking]

---

These patterns reflect common associations across modalities, contributing to spurious correlations within LMMs during pretraining.

## A.3 BENCHMARK DATA STATISTICS

Tab. 6 and Tab. 7 provides an overview of the subcategories within our benchmark framework, divided into two primary contributors to hallucinations: overreliance on unimodal priors and spurious inter-modality correlations.

In Tab. 6, "Overreliance on Unimodal Priors" explores how LMMs tend to over-focus on a single modality, leading to hallucinations. It assesses Visual Dominance and Audio Dominance based on their combined visual and audio inputs at different granularities (event- and object-level), and Language Dominance with visual-only input across both object- and event-levels.

The Tab. 7, "Spurious Inter-modality Correlations," examines the erroneous associations between modalities that lead to hallucinations. The Visual-Language and Audio-Language correlations are assessed at object- and event-level granularities, and event-level only, respectively. The Visual-Audio-Language subcategory captures the more complex interplay between visual and audio content at both object- and event-level granularities.

Figure 4 presents the statistics of object and event frequencies in the probing questions within the benchmark, categorized across different scenarios. Subfigures 4a 4b highlight the top 10 most frequently queried existent and non-existent objects, respectively, demonstrating the distribution of visual objects targeted in object-level probing. Subfigures 4c 4d display the top 10 existent and non-existent visual events, offering insight into the event-level queries specific to visual content. Finally, subfigures 4e 4f show the top 10 most common audio events, both existent and non-existent, which are critical for understanding how models handle audio-centric scenarios. These statistics ensure a balanced and comprehensive distribution of queries across different modalities and types of events or objects, facilitating robust evaluation of multimodal models.

## A.4 BENCHMARK DATA CONSTRUCTION DETAILS

The benchmark is designed to evaluate hallucination scenarios across multiple modalities, targeting specific LMM tendencies such as overreliance on individual modalities and spurious inter-modality correlations. It comprises video, audio, and textual inputs with probing questions aimed at assessing the presence or absence of objects or events in these modalities. Precise annotation is employed to ensure a thorough evaluation of LMM performance in multimodal contexts.

### A.4.1 CONSTRUCTING QUERIES FOR OVERRELIANCE ON UNIMODAL PRIORS

To assess how LMMs may excessively depend on a single modality (visual, audio, or language), we construct targeted probing queries that test this overreliance while potentially neglecting complementary information.

**Visual Dominance**   The Visual Dominance subcategory examines the extent to which LMMs overrely on visual content, potentially leading to hallucinated sound events that are often associated with visual objects. All probing questions focus on audio events. For queries about existent sound events, the ground truth "yes" is derived from direct human annotation. To identify non-existent sound events, we use the AudioCaps dataset (Kim et al., 2019), which provides short captions describing the audio track. Objects associated with these audio events are extracted using LLaMA3 (Dubey et al., 2024) from the audio caption, while visual objects are identified from video frames using InternVL2 (Chen et al., 2023). Samples where visual objects do not correspond to any audio content are filtered and manually verified, with the ground truth set to "no." All raw video-audio pairs are sourced from AudioCaps.

**Audio Dominance**   The Audio Dominance subcategory explores how LMMs may over-rely on audio cues, leading to hallucinations of visual content. Here, questions probe the presence of visual objects. For existent objects, the ground truth "yes" is annotated manually. To find non-existent objects, we filter samples where the objects indicated by audio cues are not visually present in the video. These samples undergo manual review to ensure accurate annotation, with the ground truth as "no." All raw video-audio pairs are also sourced from AudioCaps.

**Language Dominance**   The Language Dominance subcategory targets hallucinations caused by the LMMs' dependence on language priors from pretraining corpora. This category focuses on common-sense events and object attributes. We manually define sets of typical events (e.g., "fish swim in water") and object characteristics (e.g., "yellow banana"). Videos depicting anti-common-sense scenarios (e.g., "fish fly in the air," "black banana") are then collected from YouTube. For queries probing existent content, the ground truth "yes" corresponds to the anti-common-sense object/event depicted in the video. Conversely, non-existent content queries, which are the common-sense versions that do not match the video, have the ground truth "no."

Each subcategory includes 200 video-audio or video-only samples, each accompanied by two probing questions: one querying an existent object/event ("yes"), and another probing a non-existent one ("no"). For subcategories containing both object- and event-level probing, the dataset is balanced with equal numbers of object- and event-level queries.

### A.4.2 CONSTRUCTING QUERIES FOR SPURIOUS INTER-MODALITY CORRELATIONS

This section outlines the construction of queries targeting *Spurious Inter-modality Correlations*, where hallucinations arise from misleading associations between different modalities learned during pretraining. These correlations are probed at both object- and event-level granularities.

**Visual-Language Spurious Correlations**   Visual-Language spurious correlations occur when LMMs hallucinate visual objects due to associations learned from patterns in video-caption pretraining data. Queries in this subcategory are developed based on two factors: global appearance frequencies and co-occurrence patterns within the data.

- *Object-level* queries are derived from two sources: (i) global appearance frequencies, where the model is asked about frequent objects that are absent in the video (e.g., "Did you see a tree in the video?" when no tree is present), and (ii) co-occurrence patterns, where queries target non-existent objects that are often seen alongside other objects in the pretraining data (e.g., "Did you see a phone in the video?" when a human is present but no phone).

- *Event-level* queries similarly explore global appearance frequencies by probing events that frequently occur in pretraining data but are not present in the video. For co-occurrence patterns, event-level queries are designed around subject-fixed action-object pairs, such as "Did you see a person using a phone in the video?" when the person is engaged in a different action like walking.

Both global frequencies and co-occurrence data are extracted from the large-scale video-caption pretraining dataset WebVid10M. Probing samples are curated accordingly from the same source.

**Audio-Language Spurious Correlations**   This subcategory assesses correlations learned from audio-caption pretraining, leading to potential hallucinations of audio events based on their appearance or co-occurrence in the training data. Due to the temporal nature of audio, all queries are event-level.

- *Event-level* queries focus on global appearance frequencies, probing for hallucinated audio events that are common in the pretraining data but absent from the audio track (e.g., "Did you hear a dog barking?" when no such sound exists). Co-occurrence queries involve subject-fixed action-object pairs, targeting frequently co-occurring events (e.g., "Did you hear a dog barking?" when only dog whimpering is present).

The dataset Auto-acd is used for constructing these queries, ensuring a balanced representation of global appearance and co-occurrence-based patterns.

**Visual-Audio-Language Spurious Correlations**   The Visual-Audio-Language subcategory captures cross-modal hallucinations, where visual objects are hallucinated based on audio cues, and vice versa.

- *Event-level* queries test for non-existent audio events that are frequently co-occurred with visual objects in training data (e.g., "Did you hear car revving?" when a human is visible without any car sound).

- *Object-level* queries target visual objects that are hallucinated based on associated sound events (e.g., "Did you see a tree in the video?" when bird chirping is present without any tree visible).

The co-occurrence frequencies between visual objects and audio events are computed using the Auto-acd dataset, with the visual and audio content reviewed and annotated by human reviewers. Queries are evenly split between probing audio events and visual objects.

For all subcategories, there is a balance between object-level and event-level queries. Additionally, the samples constructed from global appearance frequencies and co-occurrence patterns are evenly distributed.

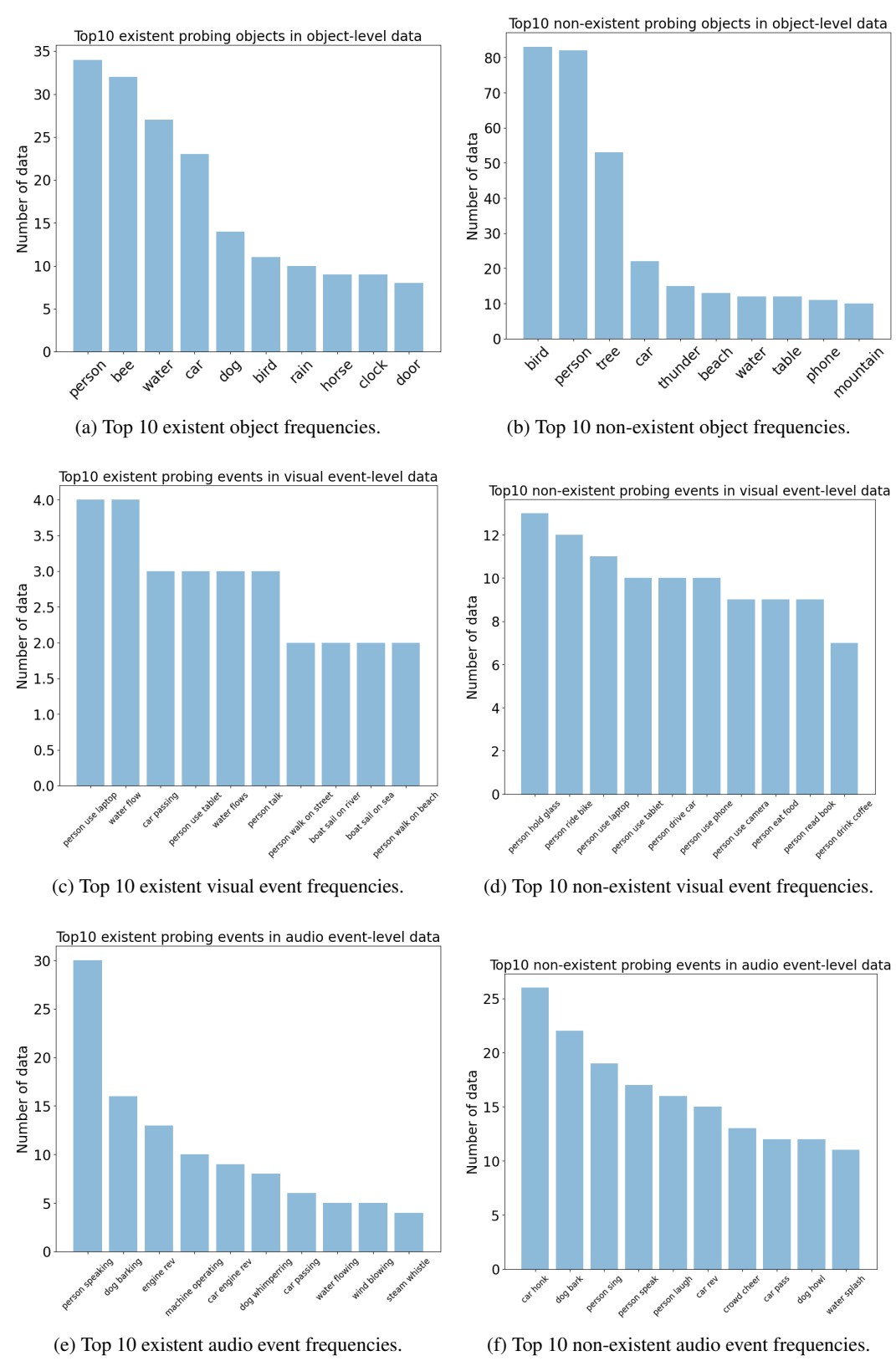

(a) Top 10 existent object frequencies.

(b) Top 10 non-existent object frequencies.

(c) Top 10 existent visual event frequencies.

(d) Top 10 non-existent visual event frequencies.

(e) Top 10 existent audio event frequencies.

(f) Top 10 non-existent audio event frequencies.

Figure 4: Statistics of object and event frequencies in our probing questions.

