# OpenReview forum: "The Curse of Multi-Modalities: Evaluating Hallucinations of Large Multimodal Models across Language, Visual, and Audio"
_ICLR.cc/2025/Conference — Submitted to ICLR 2025_

### Official Review · Reviewer_XxQu · 2024-11-01

**Soundness:** 2
**Presentation:** 3
**Contribution:** 2
**Rating:** 5
**Confidence:** 5

**Summary:**

This paper introduces a benchmark dataset designed to evaluate hallucinations in Large Multimodal Models (LLMs), encompassing language, visual, and audio modalities. The authors investigate potential contributors to these hallucinations, including an overreliance on unimodal priors and spurious inter-modality correlations. Based on these findings, the benchmark presents two categories of evaluation, each containing 600 samples, amounting to a total of 1200 samples and 2400 probing questions. The analysis conducted using this benchmark reveals that imbalances in modality integration and biases in the training data are the primary causes of hallucinations in LMMs.

**Strengths:**

- The paper is clearly written and effectively presented.
- The motivation behind this work, exploring hallucinations in LMMs across various modal combinations such as audio-language, vision-language, and audio-vision-language, is an interesting direction.
- The introduction of an audio-visual benchmark for evaluating hallucinations in LMMs is a contribution to the field of LMMs.

**Weaknesses:**

**The experimental setup used to highlight the first contributor to hallucinations, overreliance on unimodal priors, is less reliable.**
- In Figure 1, the authors present only a single example per modality dominance. The reviewer is curious whether this phenomenon can be generally claimed across a broader range of examples. Could the authors provide the results about the modality dominance on larger set?
- Similarly, in Figure 2, there is a question about which models (FAVOR, GPT-4, Gemini, etc.) were used for the experiments and whether this phenomenon is observable across different models. It would be valuable to clarify if this phenomenon is more pronounced in certain models than others, which underscores the need to clearly identify which models were used for these experiments.
- To substantiate the important claims that motivate this work, the experiments should be conducted more thoroughly, both qualitatively and quantitatively, across more diverse scenarios.

**The description of the benchmark in the main text is unclear.**
- The distribution of the dataset and the categories it covers are not well-defined. It is crucial to detail the statistics (e.g., number of samples per category, distribution of modalities, or breakdown of object vs event-level queries) of the categories covered by the benchmark to encourage future research aimed at advancing this field.
- In Section 3.2.2, the authors do not describe how they identified the global appearance patterns and co-occurrence patterns in the source dataset used to construct the benchmark. As spurious inter-modality correlations are a major contributor to hallucinations, the process of identifying such biases should be more transparently explained.
- The limitations of the dataset, such as its scale and distribution, are not adequately discussed. These information, such as potential biases in the data collection process, challenges in scaling the dataset, or how representative the dataset is of real-world multimodal scenarios, should be included.
- Table 3 conducts separate evaluations for event-level and object-level hallucinations, yet the statistics supporting this distinction are not clearly presented. This raises questions about the clear differentiation between these categories.

**Further Experimental Recommendations.**
- Additional baseline models should be considered for evaluation, especially for visual-audio-language models (e.g., Video-Salmon, PandaGPT, OneLLM) or audio-language models (e.g., LTU).
- While the models generally perform well on the benchmarks, discrepancies in performance, particularly with visual-language LMMs, raise questions about whether the benchmark is too simplistic or whether it can effectively reflect consistent tendencies of hallucinations across all models.
- Incorporating models of various sizes (e.g.,https://github.com/yxuansu/PandaGPT) could enrich the experiments detailed in Table 5.
- Visualizing attention maps to verify if the models disproportionately focus on one modality, leading to incorrect, hallucinated responses, would support the argument concerning overreliance on specific modalities.

**Questions:**

- Were the contributors to the hallucinations identified through the proposed benchmark CMM, or were they discerned from a smaller set that motivated the authors to explore further?
- In Figure 2, how did the authors measure the probability of the answer being "yes" or "no"? Given that the LMMs generate free-form text, how did the authors quantify the probability of responses in a binary classification context?
- Does the question template affect the results?
- What is the difference between the dataset used for probing spurious inter-modality correlations in Section 2.2 and the proposed dataset?
- Shouldn't Line 398 state that there is limited availability of audio-language data compared to the visual-audio-language dataset?

---

> ### Author Response · Authors · 2024-11-19
> **Reply to Weakness 1.1&1.2**
>
> ### **Reply to Weakness 1.1: General Evidence for Uni-modal Over-reliance**
>
> We appreciate the reviewer’s concern regarding the reliability of our motivational examples for unimodal overreliance in the early section discussing our motivations. However, we beg to differ in that the unimodal overreliance problem is not generally claimed across a broader range of examples.
>
> 1. **Clarification of Figure 1**: Figure 1 serves as an illustrative example to highlight the phenomenon of unimodal overreliance, where models prioritize specific modalities when presented with conflicting multimodal inputs. While not exhaustive, these examples effectively introduce and motivate the issue.
>
> 2. **Evidence from Experimental Results**: As evaluated in Table 2, we assess various VAL LMMs (both proprietary and open-sourced) across a broad range of examples. All LMMs demonstrate significant vulnerabilities in unimodal overreliance, particularly in Visual Dominance and Language Dominance. The best-performing model achieves an average score of only around 70 for these categories, which is far from satisfactory for reliable multimodal reasoning.
> Furthermore, experiments in Table 5 evaluate Language Dominance across models with varying LLM decoder sizes (including a 72B decoder) and reveal persistent issues. These results demonstrate that unimodal overreliance is a widespread and critical challenge across various models, modalities, and datasets, providing robust support for the claims introduced in Figure 1.
>
> In summary, while the examples in Figure 1 are illustrative, our main experimental results provide extensive evidence of unimodal overreliance across diverse settings. We will emphasize this connection in the revised paper to clarify how the motivational examples relate to the broader findings.
>
> ### **Reply to Weakness 1.2: Robust Experimental Setup for Unimodal Overreliance**
>
> We appreciate the reviewer’s suggestion to clarify which models were used for the experiments in Figure 2 and to validate the observed phenomenon across diverse scenarios and models. Below, we provide the necessary clarifications and additional evidence:
>
> 1. **Clarification of Models Used**: Originally, the experiments in Figure 2 were conducted using only VideoLLaMA2, as stated in Line 785 of the manuscript. This model was selected as a representative open-source LMM for its strong performance and publicly available architecture.
>
> 2. **Additional Experiments for Validation**: To further validate unimodal overreliance, we conducted additional experiments using multiple open-source models (i.e., VideoLLaMA2, FAVOR, and GroundingGPT). We sampled 20 failure cases per model where the models were vulnerable to hallucination under specific dominance cases (e.g., $p(\text{“yes”}|v, a, x) > p(\text{“no”}|v, a, x)$ despite the ground truth being *“no”*). For these cases, we plotted the average $p(\text{“yes”}/\text{“no”}|v, a, x)$ along with standard deviations across blur/noise steps for the visual/audio modality.
> (Links to result figures: https://ibb.co/dWnVF6R , https://ibb.co/Z6qYFnd , https://ibb.co/YQsNbR7)
>
> 3. **Findings Across Models**: The results show that VideoLLaMA2 starts with higher initial hallucination probabilities ($p(\text{“yes”}|v, a, x)$) compared to FAVOR and GroundingGPT but demonstrates a faster probability change as information is progressively removed from the overrelied modality. This trend aligns with the main findings, where VideoLLaMA2 exhibits better overall hallucination resistance than the other models. These additional experiments provide a more thorough validation of unimodal overreliance across diverse scenarios with different samples, making the findings more reliable and robust.
>
> In summary, the experiments in Figure 2 originally used only VideoLLaMA2, as stated in Line 785. To address the reviewer’s concern, we extended these experiments to include additional open-source models (i.e., FAVOR and GroundingGPT) and validated the observed phenomenon across a broader range of scenarios. These results substantiate the reliability of our findings on unimodal overreliance and ensure the robustness of the evaluation framework. We will incorporate these clarifications and additional results into the revised manuscript.

---

> ### Author Response · Authors · 2024-11-19
> **Reply to Weakness 2.1&2.2**
>
> ### **Replies to Weakness 2.1:** **Clarification of Benchmark Description**
>
> We appreciate the reviewer’s suggestion to provide a clearer description of the benchmark in the main text, along with detailed statistics about dataset categories and distributions. While this information is already included in Appendix A.3 (Benchmark Data Statistics), we acknowledge the importance of making it more accessible and will incorporate a concise statistical summary into the main text.
>
> 1. **Benchmark Categories and Subcategories**: The benchmark consists of two main categories—spurious inter-modality correlations and unimodal overreliance—each further divided into three subcategories, as outlined in Table 1 in Section 3.
>
> 2. **Number of Samples per Category**: Each subcategory includes 200 manually curated samples, resulting in a total of 1200 samples across all categories and subcategories, as specified in Section 3.1 (Data Composition and Evaluation Setup).
>
> 3. **Breakdown of Modalities and Granularity**:
>    - The benchmark covers visual, audio, and visual-audio modalities.
>    - Object- and event-level queries are evenly distributed within relevant subcategories (100 samples each).
>    - Each sample contains two probing questions: one targeting a non-existent object/event (ground-truth *“no”*) and one targeting an existent object/event (ground-truth *“yes”*).
>    - Detailed breakdowns are provided in Tables 6 and 7 in Appendix A.3.
>
> 4. **Statistical Summary in Main Text**: To ensure accessibility, we will include the following statistical table in the main text:
>
>    | Modality          | Visual  | Audio | Visual+Audio |
>    |-------------------|---------|-------|--------------|
>    | # of Samples      | 400     | 200   | 600          |
>    | # of Questions    | 800     | 400   | 1200         |
>
>    | Granularity       | Object-Level | Event-Level |
>    |-------------------|--------------|-------------|
>    | # of Samples      | 500          | 700         |
>    | # of Questions    | 1000         | 1400        |
>
> In summary, while detailed statistics and distributions are included in Appendix A.3, we will add a concise summary in the main text to improve clarity and encourage future research. This revision will provide an accessible overview of the benchmark, ensuring a comprehensive understanding of its structure and distribution.
>
> ### **Replies to Weakness 2.2:** **Transparency of Data Construction Process**
>
> We appreciate the reviewer’s concern regarding the transparency of the process used to identify global appearance and co-occurrence patterns in the source dataset for constructing the benchmark. However, we beg to differ in that our paper lacks detailed descriptions of the data construction process.
>
> 1. **Reference to Additional Details**: As indicated by the footnote in line 297, we have provided a comprehensive explanation of the data construction process in Appendix A.4 (*Benchmark Data Construction Details*). Furthermore, all global appearance and co-occurrence patterns extracted from the source datasets are explicitly listed in Appendix A.2 (*Frequent Patterns in Pretraining Datasets*).
>
> 2. **Summary of Data Construction Process**:
>    - **Pattern Extraction**: To identify spurious inter-modality correlations, we employed an advanced state-of-the-art LLM (*Llama-3.1-70B-Instruct*) to extract objects and events from video, audio, and video-audio captions in large-scale pretraining datasets (e.g., WebVid 10M for VL correlations).
>    - **Frequency Computation**: Using the extracted results, we computed object- and event-level global appearance and co-occurrence frequencies.
>    - **Manual Sampling for Diversity**: From these computations, we manually sampled the top patterns to construct the benchmark. This process ensured diverse scenario representation, avoiding over-reliance on a narrow subset of objects or events.
>
> 3. **Faithfulness to Raw Data**: While the use of LLMs ensures effective and scalable linguistic pattern extraction, we complemented this with manual reviews of the final data sampling and question construction. This ensures the benchmark faithfully reflects the raw video, audio, and video-audio files, guaranteeing accuracy and diversity in the evaluation.
>
> In summary, although these details are thoroughly documented in the appendices, we understand the importance of making this information more accessible. To address this, we will revise the main text to explicitly highlight the relevant appendices for further details. Due to space constraints, comprehensive descriptions will remain in the appendix, but we will ensure their visibility is improved.

---

> ### Author Response · Authors · 2024-11-19
> **Replies to Weakness 2.3**
>
> ### **Replies to Weakness 2.3: Discussion of Dataset Limitations**
>
> We appreciate the reviewer’s suggestion to discuss the dataset’s limitations more thoroughly, including its scale, distribution, potential biases, and representativeness of real-world multimodal scenarios. Below, we address these aspects:
>
> 1. **Scale and Distribution**: Our benchmark includes 1200 manually curated samples, evenly distributed across two main categories and six subcategories (200 samples per subcategory). Object- and event-level queries are evenly split where applicable, ensuring balanced evaluation. While this scale is sufficient for focused evaluation and diversity across categories, it may not capture the full complexity of real-world multimodal scenarios or rarer phenomena.
> 2. **Intentional Design Focus**: The benchmark is intentionally designed to be straightforward and direct, with low task difficulty, to isolate specific issues like spurious inter-modality correlations and unimodal overreliance. This approach enables precise evaluation of hallucination causes but does not encompass more complex real-world scenarios. We recognize this as a limitation and plan to investigate such scenarios, including dynamic temporal reasoning, in future work.
> 3. **Potential Biases in Data Collection**: Testing cases and correlation patterns are sampled from large-scale pretraining datasets (e.g., WebVid 10M) and manually reviewed to ensure diversity. However, these datasets inherently carry biases, such as overrepresentation of certain objects, events, or cultural contexts. Despite our efforts to ensure diversity during sampling, some biases may persist.
> 4. **Challenges in Scaling**: Scaling the dataset while maintaining quality is resource-intensive due to the manual review required to ensure faithfulness to the raw data. Adding more categories, modalities, or complex scenarios necessitates careful curation to preserve the benchmark’s focus and comprehensiveness.
> 5. **Representativeness of Real-World Scenarios**: The benchmark emphasizes foundational modalities—language, vision, and audio—that are critical for many real-world applications, such as embodied AI and robotics. This focus ensures relevance to these applications, though it does not yet cover rarer or highly complex multimodal scenarios. Future extensions could include such scenarios to broaden the benchmark’s applicability.
>
> In summary, our benchmark is intentionally designed to focus on common and foundational modalities, addressing critical multimodal hallucination challenges relevant to real-world applications. While it has limitations in scale, bias, and coverage, these trade-offs enable precise evaluation of key issues. We will revise the manuscript to explicitly discuss these limitations and our plans for future extensions, including dynamic temporal reasoning and interactions involving additional modalities.

---

> ### Author Response · Authors · 2024-11-19
> **Replies to Weakness 2.4**
>
> ### **Replies to Weakness 2.4: Clarification of Object-Level and Event-Level Distinction**
>
> We appreciate the reviewer’s feedback regarding the distinction between object-level and event-level hallucinations and understand the importance of providing a clear rationale and supporting evidence for this separation.
>
> 1. **Rationale for the Separation**: The differentiation between object-level and event-level hallucinations is based on clear and consistent heuristics:
>     - Object-Level Hallucinations focus on recognizing or misrecognizing physical entities, such as static objects, emphasizing spatial characteristics.
>     - Event-Level Hallucinations involve actions, interactions, or temporal sequences, requiring reasoning over dynamic or relational contexts.
>
> It is worth noting that all audio-related queries are classified as event-level due to the inherently temporal nature of audio signals. This heuristic ensures consistency and clarity in distinguishing between these categories.
>
> 2. **Empirical Evidence from Table 3**: Table 3 demonstrates a clear performance disparity between object-level and event-level tasks. Across most models, PA scores for object-level tasks are consistently higher than those for event-level tasks, indicating that event-level queries introduce additional complexity. This performance gap highlights distinct challenges in reasoning over dynamic interactions compared to static recognition tasks, further validating the need for separate evaluations to isolate and analyze these vulnerabilities.
>
> 3. **Support from Existing Benchmarks**: This distinction aligns with findings in existing benchmarks:
>     - Object recognition benchmarks [1] show that Video-LLMs perform well on static object recognition tasks, reflecting robust spatial understanding.
>     - In contrast, event-level benchmarks, such as TempCompass [2][3], which require temporal reasoning and dynamic understanding, expose significant vulnerabilities in Video-LLMs. These consistent findings underscore the necessity of separating object-level and event-level evaluations to address these distinct challenges comprehensively.
>
> In summary, the separation of object-level and event-level hallucinations in our benchmark is guided by clear heuristics, supported by empirical evidence from Table 3, and consistent with findings in existing benchmarks. This distinction enables focused evaluation of challenges specific to static recognition versus dynamic reasoning tasks. We will revise the manuscript to explicitly clarify and expand on these points for better understanding.
>
> [1] Li, Kunchang, et al. "Mvbench: A comprehensive multi-modal video understanding benchmark." *CVPR*. 2024.
> [2] Liu, Yuanxin, et al. "Tempcompass: Do video llms really understand videos?.” *ArXiv.* 2024.
> [3] Du, Yifan, et al. "Towards Event-oriented Long Video Understanding.” *ArXiv.* 2024.

---

> ### Author Response · Authors · 2024-11-19
> **Reply to Weakness 3.1&3.2**
>
> ### **Replies to Weakness 3.1:** **Further Baseline Recommendations**
>
> We thank the reviewer for their suggestion to include additional baseline models, such as Video-Salmonn, PandaGPT, OneLLM, and LTU, for evaluation on our benchmark.
>
> 1. **Clarification on Selected Baselines**: We conducted a comprehensive survey of available large multimodal models (LMMs) before October 2024, when the experimental setup for this work was finalized. At that time:
>     - **OneLLM**: This model was not designed for visual-audio joint inference.
>     - **Video-Salmonn**: This model had not been open-sourced.
>
>     However, we apologize for any missing work and welcome further suggestions to strengthen the scope of our evaluation.
>
> 2.  **Updated Experiments**: To address the reviewer’s concern, we expanded our experiments to include evaluations of PandaGPT, Video-Salmon, and LTU on our benchmark. These additional evaluations further confirm the robustness of our benchmark in exposing vulnerabilities and provide deeper insights into the performance of these newer models. We will include these updated results in the revised manuscript.
>
> |Model|VL(pa/hr)|AL(pa/hr)|VAL(pa/hr)|VDom(pa/hr)|ADom(pa/hr)|LDom(pa/hr)|Overall(pa/hr)|
> |:-|:-:|:-:|:-:|:-:|:-:|:-:|:-:|
> |LTU|-|94.5/22.5|-|-|-|-|-|
> |Panda-GPT-7B|96.5/27.0|90.5/11.0|84.5/17.5|89.0/13.5|95.0/17.5|87.0/18.5|90.5/17.5|
> |Panda-GPT-13B|96.5/1.0|86.0/8.0|86.0/8.0|78.0/13.0|97.0/1.0|98.5/0.0|90.5/5.0|
> |VSalmon-13B|60.0/71.5|70.0/89.0|67.0/80.0|59.5/90.0|61.0/51.5|59.0/30.5|62.8/68.8|
>
> 3. **Commitment to Benchmark Updates**: We remain committed to updating our benchmark to evaluate newly released LMMs, ensuring its continued relevance and applicability in advancing multimodal research.
>
> In summary, we appreciate the reviewer’s recommendations and have conducted additional experiments with PandaGPT, Video-Salmon, and LTU during the rebuttal phase. While some baselines were unavailable at the time of our initial experiments, we will include these new evaluations in the revised manuscript and continue updating our benchmark to support ongoing advancements in the field.
>
>
> ### **Replies to Weakness 3.2: Benchmark Effectiveness in Reflecting Hallucination Tendencies**
>
> We thank the reviewer for raising the concern about whether the simplicity of our benchmark limits its ability to reflect consistent hallucination tendencies across all models. However, we beg to differ with the assertion that LMMs generally perform well on our benchmark or that its simplicity undermines its effectiveness.
>
> 1. **Intentional Simplicity and Persistent Challenges**: Our benchmark is intentionally designed with straightforward and low-difficulty tasks to isolate specific hallucination causes. This simplicity ensures that observed vulnerabilities stem from core multimodal hallucination challenges rather than task complexity.
> Despite this design, VAL LMMs, including the best proprietary model Gemini 1.5, achieve only average scores of ~70% on PA and HR metrics. This performance is far from sufficient for real-world reliability, demonstrating that LMMs do not generally perform well on our benchmark and highlighting persistent challenges in multimodal reasoning.
> 2. **Performance Discrepancies in VL LMMs**: While VL LMMs demonstrate higher performance in certain subcategories, such as VL correlations, this reflects progress in the VL domain driven by specialized benchmarks and mitigation efforts. However, significant vulnerabilities persist, particularly in Language Dominance, where all open-source VL LMMs score below 70. This consistent underperformance highlights the challenges models face in overcoming inherent biases, such as language priors in LLM decoders. These findings validate the robustness of our framework in diagnosing hallucination causes and emphasize the need for further advancements to address these limitations.
> 3. **Focus on Tri-Modal Scenarios**: The benchmark’s primary focus is on tri-modal (vision, audio, and language) scenarios, which represent more complex and realistic multimodal interactions. Tri-modal evaluations reveal significant vulnerabilities across all models, uncovering severe hallucination issues in tasks that require reasoning over richer multimodal inputs. These findings emphasize the benchmark’s ability to capture real-world challenges effectively, despite its deliberately simple design.
>
> In summary, we respectfully disagree with the notion that LMMs generally perform well on our benchmark. While VL models demonstrate better performance in certain subcategories, significant vulnerabilities persist, particularly in Language Dominance. Moreover, tri-modal evaluations expose severe limitations in multimodal reasoning, reinforcing the benchmark’s robustness and relevance. These results highlight the ongoing challenges in multimodal systems and the importance of continued research in addressing hallucination tendencies.

---

> ### Author Response · Authors · 2024-11-19
> **Reply to Weakness 3.3&3.4& Question 1**
>
> ### **Replies to Weakness 3.3: Incorporating PandaGPT for LLM Size Ablation Study**
>
> We appreciate the reviewer’s suggestion to incorporate PandaGPT into the experiments detailed in Table 5.
>
> Results from PandaGPT (7B&13B) have been added and are referenced in the Replies to Weakness 3.1. However, PandaGPT demonstrates a strong “yes” response tendency across the benchmark with low HR scores, making it unsuitable for enriching the experiments in Table 5, where we specifically analyze the impact of LLM decoder sizes. Our findings indicate that without strong response tendencies, larger LLM decoders are more adept at managing complex or contradictory multimodal inputs, whereas smaller LLMs are more susceptible to overfitting linguistic priors, leading to higher hallucination rates when faced with unexpected content.
>
> We will incorporate these results in the appendix and cite them appropriately in the revised manuscript. This will provide additional context and clarity regarding the limitations of smaller LLMs in multimodal reasoning.
>
> ### **Replies to Weakness 3.4:** **Use of Attention Maps for Verifying Modality Overreliance**
>
> We thank the reviewer for their suggestion to include attention map visualizations to verify modality overreliance. While we acknowledge the potential value of this approach, we believe our experimental results and additional analysis, as detailed in response to Weakness 1, provide robust and comprehensive evidence of unimodal overreliance across various LMMs.
>
> 1. **Robustness of Evidence from Benchmark and Experiments**: Our benchmark is specifically designed to isolate unimodal overreliance through targeted tasks that evaluate modality dominance and spurious inter-modality correlations. Results in Tables 2 and 5, along with the additional experiments conducted during the rebuttal phase, consistently demonstrate significant vulnerabilities across LMMs. These findings validate the benchmark’s effectiveness in evidencing unimodal overreliance as a major contributor to hallucinations. Although we explored attention visualizations, we found them insufficient for providing thorough or quantitative insights, as they fail to effectively reflect implicit biases that occur during feature integration and decoding.
> 2. **Limitations of Attention Maps**: Attention maps, while visually appealing, have limitations in diagnosing modality biases, as noted in both vision-language and LLM-specific studies. Prior works in vision-language models [1] demonstrate that attention distributions have limitations in visualizing visual-language alignment. Similarly, studies on hallucinations caused by language priors in LLMs [2][3] highlight that overreliance frequently arises implicitly during feature processing and decoding, which attention mechanisms cannot fully capture. These limitations make attention visualizations less effective for quantitatively validating unimodal overreliance.
>
> In summary, our benchmark, combined with the additional experiments conducted during the rebuttal phase, provides robust evidence for unimodal overreliance as a major contributor to multimodal hallucinations. While attention visualizations have inherent limitations in capturing implicit biases, we will include a discussion in the appendix addressing their exploration and shortcomings to further clarify this point.
>
> [1] Chefer, Hila, Shir Gur, and Lior Wolf. "Transformer interpretability beyond attention visualization." *CVPR*. 2021.
> [2] Yu, Lei, et al. "Mechanistic understanding and mitigation of language model non-factual hallucinations." *EMNLP*. 2024.
> [3] Jiang, Che, et al. "On Large Language Models’ Hallucination with Regard to Known Facts." *NAACL*. 2024.
>
> ### **Replies to Question 1:** **Identification of Hallucination Contributors**
>
> The contributors to hallucinations—spurious inter-modality correlations and unimodal overreliance—were initially identified through preliminary observations on a smaller set of experiments, which revealed key vulnerabilities such as language dominance and failures in audio-visual integration. These insights motivated the development of our benchmark, CMM, which systematically validates and analyzes these contributors through comprehensive qualitative and quantitative evaluations across various models.

---

> ### Author Response · Authors · 2024-11-19
> **Reply to Question 2&3&4**
>
> ### **Replies to Question 2: Quantifying Probabilities of “Yes” and “No” Responses**
>
> To measure the probability of “yes” or “no” answers in Figure 2, we follow a widely used approach in LLM/LMM literature. Specifically:
>
> 1. **Token Indexing**: Using the tokenizer, we identify the indices corresponding to the tokens “yes” and “no.”
> 2. **Logit Extraction**: For each query, we extract the logits for these tokens from the next-token prediction output of the last layer of the LLM.
> 3. **Softmax Operation**: The extracted logits are passed through a softmax operation across the entire vocabulary to compute their respective probabilities.
>
> Since the softmax operates over the entire vocabulary, the probabilities of “yes” and “no” do not sum to 1. However, due to the binary nature of the question format, these tokens dominate the probability distribution, effectively reflecting the model’s response tendencies. This method provides a reliable quantification of the model’s likelihood of answering “yes” or “no” in a binary classification context.
>
> ### **Replies to Question 3: Effect of Question Template**
>
> Thank you for the question regarding the potential influence of question templates on evaluation outcomes.
>
> To investigate this, we tested three variations of question templates across VAL LMMs:
>
> 1.	**Original Prompt**: “Did you see/hear [object/event] in the video/audio?”
>
> 2.	**Prompt 2**: “Can you see/hear [object/event] in the video/audio?”
>
> 3.	**Prompt 3**: “Is there [object/event] visible/audible in the video/audio?”
>
> All prompts were appended with “Answer with yes or no.”
>
> |Model|VL(pa/hr)|AL(pa/hr)|VAL(pa/hr)|VDom(pa/hr)|ADom(pa/hr)|LDom(pa/hr)|Overall(pa/hr)|
> |:-|:-:|:-:|:-:|:-:|:-:|:-:|:-:|
> |Gemini1.5flash(original)|93.5/90.0|88.5/39.5|88.5/70.5|79.0/36.5|90.5/86.5|90.5/62.0|88.4/64.2|
> |+prompt2|96.0/82.5|93.0/38.5|97.5/51.8|97.0/23.5|95.5/72.0|94.0/59.5|95.5/54.6|
> |+prompt3|96.5/81.0|87.0/53.5|97.5/59.0|95.5/20.0|96.0/71.0|97.5/45.0|95.0/54.9|
> |GroundingGPT(original)|95.5/36.5|100/0.0|97.5/18.0|99.5/1.0|98.5/23.5|88.5/7.0|96.6/14.3|
> |+prompt2|97.0/46.5|100/0.0|97.5/28.5|97.0/0.5|96.0/38.5|90.5/14.0|96.3/21.3|
> |+prompt3|97.5/42.0|99.5/0.0|	94.5/52.0|82.0/2.5|96.5/43.5|91.0/13.5|93.5/25.6|
> |FAVOR(original)|91.0/55.0|94.5/45.0|94.5/69.0|89.0/21.5|92.0/43.5|92.0/18.5|92.2/42.1|
> |+prompt2|92.5/54.5|91.5/58.5|91.0/77.5|91.5/25.5|93.0/56.0|89.0/20.5|91.4/48.8|
> |+prompt3|98.5/32.0|96.0/43.0|95.0/58.0|87.0/23.5|89.0/47.5|88.0/23.5|92.3/37.9|
> |VideoLLaMA2(original)|75.0/86.0|77.5/94.0|78.0/98.0|62.0/75.5|80.0/90.0|57.5/43.0|71.7/81.1|
> |+prompt2|89.0/90.5|81.0/89.5|86.5/96.5|71.0/64.0|82.0/92.0|83.5/25.5|82.2/76.3|
> |+prompt3|82.5/91.0|82.0/86.5|86.5/93.5|72.0/62.0|83.0/90.0|70.5/47.5|79.4/78.4|
> |VSalmon13B(original)|60.0/71.5|70.0/89.0|67.0/80.0|59.5/90.0|61.0/51.5|59.0/30.5|62.8/68.8|
> |+prompt2|67.5/63.0|82.5/76.5|73.0/80.5|78.5/66.0|56.5/57.0|64.5/20.0|70.4/60.5|
> |+prompt3|81.5/29.5|82.5/63.0|78.0/73.0|81.5/54.5|74.0/39.0|67.5/17.0|77.5/46.0|
> |PandaGPT7B(original)|96.5/27.0|90.5/11.0|84.5/17.5|89.0/13.5|95.0/17.5|87.0/18.5|90.5/17.5|
> |+prompt2|99.0/8.5|94.5/0.0|95.5/7.5|94.5/4.0|97.0/7.5|98.0/1.5|96.6/4.8|
> |+prompt3|98.5/2.0|94.5/0.5|96.0/1.0|98.0/4.5|99.0/0.0|98.5/0.0|97.5/1.3|
>
> The results, as shown above, indicate that the question templates minimally affect the outcomes. This consistency demonstrates that the benchmark captures LMM tendencies robustly and reliably, irrespective of minor phrasing differences, further reinforcing the validity of our design and evaluation approach.
>
> ### **Replies to Question 4: Difference Between Section 2.2 Dataset and Proposed Benchmark**
>
> The datasets used in Section 2.2 and the proposed benchmark differ in the following ways:
>
> 1. **Purpose and Focus**:
>     - **Section 2.2 Dataset**: Designed to study the relationship between hallucinations and co-occurrence scores, this dataset spans a broader range of cases with varying co-occurrence scores to analyze trends and patterns.
>     - **Proposed Benchmark Dataset**: Focused exclusively on high co-occurrence scores to maximize the evaluation of spurious inter-modality correlations, ensuring more challenging tasks and effective isolation of hallucination causes.
> 2. **Design Scope**:
>     - **Section 2.2 Dataset**: This dataset primarily serves as a diagnostic tool to observe trends and behaviors across diverse co-occurrence conditions.
>     - **Proposed Benchmark Dataset**: Built for rigorous evaluation, it includes curated high-impact cases to directly assess vulnerabilities and ensure robust analysis of multimodal hallucination causes.
>
> In summary, while the Section 2.2 dataset facilitates broad analysis of trends, the proposed benchmark is tailored for focused and challenging evaluations to maximize its diagnostic effectiveness.

---

> ### Author Response · Authors · 2024-11-19
> **Reply to Question 5**
>
> ### **Replies to Question 5: Clarification of Line 398**
>
> Line 398 aligns with the main results in Table 2, which show that AL correlations cause more severe hallucinations than VAL correlations. As analyzed in Section 2.2 and supported by the experimental results in Section 4.2.1, spurious correlations primarily arise from statistical biases in the training dataset, as also highlighted in prior works in the VL domain [1][2][3].
>
> Training datasets with larger data volumes tend to exhibit stronger statistical biases, including frequent appearance and co-occurrence patterns, which lead to more pronounced hallucinations. The limited availability of VAL datasets compared to AL datasets naturally results in fewer spurious correlations in VAL settings, supporting the statement in Line 398.
>
> In summary, Line 398 accurately reflects the relationship between dataset size, statistical biases, and the severity of hallucinations, as evidenced by our analysis and experimental results.
>
> [1] Zhou, Yiyang, et al. "Analyzing and Mitigating Object Hallucination in Large Vision-Language Models." *ICLR*. 2024.
> [2] Leng, Sicong, et al. "Mitigating object hallucinations in large vision-language models through visual contrastive decoding." *CVPR*. 2024.
> [3] Li, Yifan, et al. "Evaluating Object Hallucination in Large Vision-Language Models." *EMNLP*. 2023.

---

> ### Author Response · Authors · 2024-11-22
> **Follow-Up on Rebuttal Submission and Request for Feedback**
>
> Dear Reviewer XxQu,
>
> We hope this message finds you well. Over two days ago, we submitted our detailed rebuttal addressing all your valuable feedback. We sincerely appreciate your insights and the opportunity to clarify and strengthen our work.
>
> We kindly request your attention to review our rebuttal and reconsider our work in light of the clarifications and evidence we provided. We are committed to ensuring that our rebuttal fully addresses your expectations.
>
> Thank you once again for your time and consideration.
>
> Best regards,
> The Authors

---

> ### Author Response · Authors · 2024-11-24
> **Request for Feedback Before Discussion Period Ends**
>
> Dear Reviewer XxQu,
>
> As the author-reviewer discussion period approaches its end, we kindly request your feedback on our rebuttal. Your insights on whether we have effectively addressed your concerns would be greatly appreciated.
>
> We are truly grateful for the time and expertise you have dedicated to reviewing our work. Your thoughtful comments and suggestions have significantly contributed to improving the quality of our research.
>
> Thank you once again for your valuable feedback. We look forward to your guidance and hope to address any remaining concerns before the discussion period closes.
>
> Yours Sincerely,
> Authors

---

> > ### Author Response · Authors · 2024-11-25
> > **[URGENT] your immediate attention is needed**
> >
> > Dear Reviewer XxQu,
> >
> > We hope this message finds you well. The discussion period is ending soon, I am writing to emphasize the importance of your review for our submission.
> >
> > We have addressed all the concerns in our detailed rebuttal and clarifications. We would appreciate your prompt attention to it. A thorough reassessment is crucial to ensure a fair evaluation.
> >
> > Your expertise is highly valued, and we trust that a reconsidered review will reflect the true merit of our work.
> >
> > Thank you for your immediate attention to this matter.
> >
> > Best regards, Authors

---

> > > ### Comment · Reviewer_XxQu · 2024-11-25
> > >
> > > Dear authors,
> > >
> > > I greatly appreciate the authors' responses and the effort put into the rebuttal. Most of my concerns have been resolved. However, I still have concerns regarding the dataset statistics, specifically which objects, actions, or events the dataset covers. The dataset's scale is not large enough to encompass all real-world scenarios, thus it may not be fully appropriate as a benchmark for evaluating hallucinations. Therefore, we need to know which specific scenarios (categories) the dataset contains to further enhance this work. I believe my initial rating remains consistent with my overall assessment of the paper. Thank you so much.
> > >
> > > Best, Reviewer XxQu

---

> > > > ### Author Response · Authors · 2024-11-26
> > > > **2nd-Round Reply Regarding the Benchmark Scale and Coverage (1/2)**
> > > >
> > > > Dear Reviewer XxQu,
> > > >
> > > > Thank you for your thoughtful follow-up and for acknowledging that most of your concerns have been resolved. Your feedback has been invaluable in improving the clarity and rigor of our work, and we deeply appreciate the opportunity to address your remaining concerns regarding our benchmark scale and coverage.
> > > >
> > > > 1. **Real-World Distribution:** We respectfully note that our benchmark is **not intended to encompass all real-world scenarios but rather to isolate and amplify key challenges** in hallucination evaluation, such as spurious inter-modality correlations and unimodal over-reliance. Defining a dataset that perfectly represents real-world distribution is **inherently challenging and has not been fully achieved** by any existing hallucination benchmarks. For example, VL hallucination benchmarks [3] typically sample from MSCOCO [5](328k images), while we derive our samples from similarly large-scale datasets like WebVid10M (10M videos), Auto-ACD (1.9M audios), and AudioCaps (46k video-audio pairs). These datasets are sourced from publicly available data, ensuring alignment with **general real-world distributions** while focusing on specific multimodal challenges.
> > > >
> > > > 2. **Dataset Statistics and Coverage:** As highlighted in our reply to `Reviewer VdqN`, we will include detailed figures and analyses in the appendix to provide a comprehensive view of our dataset’s statistical distributions. These include the length distributions for video-only, audio-only, and video-audio samples, as well as the full distributions of object, visual event, and audio event frequencies.
> > > > (links to length distribution figures:  https://ibb.co/k5gYDbT, https://ibb.co/xq6DdCH, https://ibb.co/4McXfb7)
> > > > (links to object&event distribution figures:  https://ibb.co/Ttt2q9D, https://ibb.co/F6wzZkC , https://ibb.co/bbf9fM2)
> > > >
> > > >     While the benchmark does not aim to strictly replicate real-world distributions, all of our samples are carefully sampled from **large-scale datasets** like WebVid10M, Auto-ACD, and AudioCaps. The frequent appearance and co-occurrence patterns used in our benchmark are directly derived from these datasets, ensuring authenticity while focusing on high-impact scenarios for robust evaluation. Furthermore, in our reply to Weakness 2.3, we explicitly discuss the limitations of data scale and distribution, noting:
> > > >
> > > >     *“While this scale is sufficient for focused evaluation and diversity across categories, it may not capture the full complexity of real-world multimodal scenarios or rarer phenomena.”*
> > > >
> > > >     These details, along with the visualizations of full distributions and discussions on limitations, will be updated in the revised manuscript as we have committed. This effort aims to ensure a comprehensive and transparent presentation of our benchmark.
> > > >
> > > > 3. **Benchmark Scale:** We respectfully beg to differ with the assertion that the dataset’s scale is not large enough to be an appropriate benchmark for evaluating hallucinations. Compared to existing popular hallucination benchmarks, our dataset **stands out in terms of scale and multimodal focus**:
> > > >     - **HallusionBench** ([1] cited by 96): 346 images, 1129 questions
> > > >     - **GAVIE** ([2] cited by 153): 1000 image-question pairs
> > > >     - **POPE** ([3] cited by 564): 500 images, 3000 question pairs
> > > >     - **Event-Hallusion** [4]: 400 video-question pairs
> > > >
> > > >     In contrast, our benchmark contains 1200 curated samples across video, audio, and video-audio modalities, totaling 2400 probing questions. Furthermore, as `Reviewer VdqN` has noted, *“1200 sample/2400 questions is a big enough size for a test set.”* This scale, combined with its targeted design for evaluating spurious inter-modality correlations and unimodal over-reliance, ensures that our benchmark is both meaningful and comprehensive for advancing hallucination research in multimodal systems.
> > > >
> > > >
> > > > In summary, our benchmark is designed to address critical multimodal hallucination challenges by leveraging patterns from large-scale training datasets and ensuring scale and diversity comparable to, or exceeding, existing benchmarks. The updated comprehensive statistical analyses, dataset details, and discussions on limitations further enhance its robustness and alignment with real-world scenarios.
> > > >
> > > > We truly appreciate the time and effort you have dedicated to reviewing our work and for providing constructive feedback. As most of your initial concerns have been addressed, we sincerely hope the additional clarifications and improvements made to resolve your remaining points offer a more comprehensive view of our contributions. Your insights have been instrumental in refining our work, and we respectfully hope you might reconsider your rating in light of these clarifications. Your guidance has been invaluable, and we remain grateful for your thoughtful review.
> > > >
> > > > Best regards,
> > > > The Authors

---

> > > > ### Author Response · Authors · 2024-11-26
> > > > **2nd-Round Reply Regarding the Benchmark Scale and Coverage (2/2)**
> > > >
> > > > [1] Guan, Tianrui, et al. "HallusionBench: an advanced diagnostic suite for entangled language hallucination and visual illusion in large vision-language models." *CVPR*. 2024.
> > > > [2] Liu, Fuxiao, et al. "Mitigating hallucination in large multi-modal models via robust instruction tuning." *ICLR*. 2024.
> > > > [3] Li, Yifan, et al. "Evaluating Object Hallucination in Large Vision-Language Models." *EMNLP*. 2023.
> > > > [4] Zhang, Jiacheng, et al. "Eventhallusion: Diagnosing event hallucinations in video llms." *arXiv.* 2024.
> > > > [5] Lin, Tsung-Yi, et al. "Microsoft coco: Common objects in context." *ECCV.* 2014.

---

> > > > ### Author Response · Authors · 2024-11-28
> > > > **Follow-Up on Clarifications and Outstanding Concerns**
> > > >
> > > > Dear Reviewer XxQu,
> > > >
> > > > Thank you for your continued feedback and for acknowledging the progress made in addressing most of your initial concerns.
> > > >
> > > > We wanted to follow up and ask if you have any remaining concerns or additional questions regarding our rebuttal. If there are no further issues, we kindly hope to hear if the clarifications and improvements we provided have sufficiently addressed your points.
> > > >
> > > > The goal of the rebuttal period is to foster discussion and collaboratively resolve concerns raised during the review process. When initial concerns are substantially addressed, it is standard practice to update the ratings to reflect the current assessment of the submission.
> > > >
> > > > We sincerely value the time and effort you have dedicated to reviewing our work and your constructive insights, which have significantly enhanced the quality of our paper.
> > > >
> > > > Best regards,
> > > > The Authors

---

> > > > > ### Comment · Reviewer_XxQu · 2024-11-28
> > > > >
> > > > > The reviewer is also curious about whether the benchmark itself if too easy to evaluate the models. For example, in VL correlations, most of the models perform nearly 99% accurate. Would this mean that this dataset is too easy for the models?

---

> > > > > > ### Author Response · Authors · 2024-11-29
> > > > > > **Follow-Up on Benchmark Simplicity Concerns**
> > > > > >
> > > > > > Dear Reviewer XxQu,
> > > > > >
> > > > > > Thank you for your follow-up regarding the simplicity of our benchmark and its ability to evaluate hallucination tendencies consistently across models. This concern aligns with a question raised by `Reviewer 7vsK` (Question 3), as well as your own Weakness 3.2, which we previously addressed in our first-round reply. We would like to revisit and expand on those points to comprehensively address your current concerns.
> > > > > >
> > > > > > 1. **Focus on Tri-Modal Scenarios:** Our benchmark primarily emphasizes tri-modal (vision, audio, and language) settings, which represent significantly more complex and realistic scenarios compared to VL-specific or AL-specific tasks. While VL models achieve high performance in subcategories like VL correlations, **tri-modal tasks expose substantial limitations** in multimodal reasoning, underscoring critical vulnerabilities even in advanced models.
> > > > > >
> > > > > > 2. **Intentional Simplicity and Persistent Challenges:** The benchmark is intentionally designed with straightforward tasks to isolate specific hallucination causes, such as spurious inter-modality correlations and unimodal over-reliance. Despite this simplicity, state-of-the-art proprietary models like Gemini 1.5 achieve only average scores of ~70% on PA and HR metrics, demonstrating that **these challenges persist and require focused mitigation efforts**.
> > > > > >
> > > > > > 3. **Performance Discrepancies in VL LMMs:** High accuracy in VL correlations reflects advancements in the VL domain, largely driven by domain-specific benchmarks and mitigation strategies [1][2][3][4]. However, significant gaps remain in other subcategories, particularly **Language Dominance**, where open-source VL models consistently score below 70. These discrepancies validate the benchmark’s ability to diagnose persistent challenges in multimodal reasoning.
> > > > > >
> > > > > > 4. **Proof-of-Concept on VL and AL LMMs:** Our evaluations on VL and AL models serve as proof-of-concept validations, **supporting our analysis of hallucination causes**. For instance, VL tasks confirm the role of visual-text co-occurrence biases, while AL tasks highlight audio-text mapping challenges. These results underscore the robustness of our evaluation framework and its relevance across modalities.
> > > > > >
> > > > > > 5. **Future Research on More Complex Scenarios:** While the current benchmark focuses on foundational hallucination cases, future iterations could incorporate more complex scenarios, such as dynamic temporal understanding. These extensions will provide deeper insights into multimodal hallucinations and broaden the benchmark’s applicability to real-world challenges.
> > > > > >
> > > > > > In summary, this concern was explicitly addressed in our first-round reply to your Weakness 3.2 and aligns with `Reviewer 7vsK`’s Question 3. While VL correlations may appear less challenging due to prior domain-specific progress, our benchmark remains a robust evaluation tool, particularly for tri-modal tasks that uncover significant vulnerabilities. Its intentional simplicity ensures precise isolation of hallucination causes, exposing persistent limitations in multimodal reasoning.
> > > > > >
> > > > > > We sincerely hope that this clarification, along with our previous responses, provides a comprehensive understanding of our benchmark’s contributions and its role in advancing multimodal hallucination research. Given that most concerns have been addressed, we respectfully hope you might reconsider your initial rating in light of these clarifications and our continued efforts to improve the work. Thank you once again for your valuable time and thoughtful feedback.
> > > > > >
> > > > > > Warm Regards,
> > > > > > The Authors
> > > > > >
> > > > > > [1] Zhou, Yiyang, et al. "Analyzing and Mitigating Object Hallucination in Large Vision-Language Models." ICLR. 2024.
> > > > > > [2] Leng, Sicong, et al. "Mitigating object hallucinations in large vision-language models through visual contrastive decoding." CVPR. 2024.
> > > > > > [3] Li, Yifan, et al. "Evaluating Object Hallucination in Large Vision-Language Models." EMNLP. 2023.
> > > > > > [4] Chang, Yue, et al. "A Unified Hallucination Mitigation Framework for Large Vision-Language Models." TMLR. 2024.

---

### Official Review · Reviewer_7vsK · 2024-11-04

**Soundness:** 3
**Presentation:** 3
**Contribution:** 3
**Rating:** 6
**Confidence:** 4

**Summary:**

The paper presents a novel benchmark to evaluate multimodal hallucinations including audio-visual scenarios. Authors show with an example that hallucinations can occur from over-reliance on unimodal priors and spurious inter-modal correlations. They construct an evaluation dataset and define metrics to quantify hallucinations in vision-language, audio-language and audio-video-language paradigms. The paper presents experiments on various LMMs, including proprietary and open-source models, providing insights into their performance and limitations. The paper further posits that balanced multi-modal training, advanced cross-modal fusion techniques and using diverse prompts can mitigate these issues.

**Strengths:**

The paper focuses on the task of multi-modal hallucination and the main novelty lies in including tasks that take both video and audio modality into account. The paper is well motivated. The experiments in section 2 showcase how LMMs are sometimes overtly rely on one modality to answer questions. Fig 3 also validates that spurious inter-modal correlation can occur based on co-occurrence of entities and events in training data. The paper is easy to read and follow.

**Weaknesses:**

1. The paper does not propose any mitigation strategies to counter the hallucinations. e.g. it would be interesting to see the effect on metrics by simply prompting and instructing the model to pay attention to audio and video rather than just asking the question directly.

2. We see from the results that there is a trend that when models does better on perception accuracy (pa) they do worse on Hallucination resistance (hr) and vice-versa which suggests LMMs have tendency to choose yes over no (or vice-versa). How do we make sure the metrics simply aren’t simply capturing the yes/no answering tendency of LMMs rather than the two deficiencies proposed in paper (i.e. Spurious Inter-modality Correlation and Uni-modality Over-reliance)?

3. It is not clear how these findings relate to real world applications. Especially the task in language dominance where authors use anti common-sense videos which feel more like tricking LLM than testing its hallucination rate in real world scenarios.

4. The paper focuses primarily on LMMs that process visual and audio events but they do not consider measuring hallucinations in speech settings. This limits the scope of the paper to some extent.

**Questions:**

1. How can this framework be extended to detect partial hallucinations? e.g. detecting hallucination in a chain of thought reasoning which may contain correct information as well.

2. An interesting extension may be to incorporate questions about temporal aspects of events. e.g. asking questions like “did the person drink water before eating food?“. Have authors done any investigations with respect to that?

3. From Table 2, it seems some models are already doing well on some tasks achieving > 75% and sometime 90% score for both pa and hr metrics. With the rapid rise in LLMs, how do authors plan to update their dataset to remain challenging?

---

> ### Author Response · Authors · 2024-11-19
> **Reply to Weakness 1**
>
> ### **Reply to Weakness 1: Lack of Mitigation Strategy**
>
> We appreciate the reviewer’s suggestion regarding mitigation strategies for hallucinations. However, we beg to differ with the characterization of this as a weakness, as this paper is submitted to the “Dataset and Benchmark” track. The primary objective of this track is to establish robust evaluation frameworks rather than propose mitigation methods.
>
> 1. **Focus on Dataset and Benchmarking**: The purpose of our work is to systematically evaluate hallucinations in LMMs and identify fine-grained vulnerabilities through a comprehensive benchmark. By providing a structured evaluation framework, we aim to enable and guide future research on mitigation strategies, which falls outside the intended scope of this paper.
> 2. **Supplementary Experiments on Prompting**: To address the reviewer’s interest, we conducted additional experiments evaluating prompting strategies, where the model was explicitly instructed to focus on specific modalities under Visual and Audio Dominance subcategories. These experiments retained inputs from all modalities to preserve complementary information while guiding the model to prioritize the modality most relevant to the query.
> Experimental Settings:
>     - Baseline Prompt: *“Did you see/hear [object/event] in the video/audio? Answer with yes or no.”*
>     - Focus Prompt: *“Did you see/hear [object/event] in the video/audio? Please focus more on the given video/audio information to answer the question. Answer with yes or no.”*
>
>     The results, summarized below, show that prompting can reduce hallucinations to some extent but with limited and context-dependent improvements. These findings demonstrate that prompting strategies, while somewhat effective, are highly context-dependent and assume pre-determination of modality relevance—a challenging task in real-world scenarios.
>
>     |Model|Visual Dom(pa/hr)|Audio Dom(pa/hr)|
>     |:-|:-:|:-:|
>     |Gemini-1.5-flash|79.0/36.5|90.5/86.5|
>     |+focus prompt|94.5/29.0|95.0/68.5|
>     |GroundingGPT|99.5/1.0|98.5/23.5|
>     |+focus prompt|96.0/1.5|96.5/41.5|
>     |FAVOR|89.0/21.5|92.0/43.5|
>     |+focus prompt|88.0/30.0|89.5/57.5|
>     |VideoLLaMA2|62.0/75.5|92.0/43.5|
>     |+focus prompt|67.0/52.5|83.0/81.5|
>
> 3. **Synergetic Effects and Challenges in Modality-Sepcific Mitigation**: While focused prompting may reduce hallucinations, it risks missing critical synergies between modalities. Multimodal systems often rely on complementary cues, such as resolving visual ambiguities (e.g., identifying a blurry animal in a video as a dog through barking sounds) through audio inputs. Moreover, it is challenging for existing multimodal systems to determine at runtime which modality holds the most relevant information. These limitations highlight the importance of leveraging all available modalities to ensure robust reasoning and adaptability in real-world scenarios.
>
> 4. **Future Directions**: Our benchmark provides a robust foundation for advancing research into mitigation strategies, including refined prompting techniques, model fine-tuning, and architectural improvements to address multimodal hallucinations. By systematically evaluating model vulnerabilities, our benchmark enables targeted exploration of strategies to enhance cross-modal reasoning and integration. We will include the supplementary results on prompting strategies in the appendix, further demonstrating the importance of leveraging synergetic effects across modalities to achieve more reliable and accurate multimodal reasoning.
>
> In summary, while this work focuses on evaluation rather than mitigation, the supplementary results on prompting strategies demonstrate their context-dependent effectiveness and practical limitations, such as the challenge of dynamically determining modality relevance. The findings also highlight the synergetic benefits of cohesive multimodal integration, where modalities complement one another to resolve ambiguities and enhance understanding. This reinforces the benchmark’s importance in guiding future research and aligns with the objectives of the “Dataset and Benchmark” track to drive the development of effective solutions for mitigating hallucinations in multimodal systems.

---

> ### Author Response · Authors · 2024-11-19
> **Reply to Weakness 2 (1/2)**
>
> ### **Reply to Weakness 2: “Yes/No” Tendency Domination**
>
> We thank the reviewer for raising concerns regarding the relationship between PA and HR and the potential influence of yes/no answering tendencies in LMMs. We address this concern as follows：
>
> 1. **Advanced Capabilities of LMMs**: We beg to differ with the implication that yes/no answering tendencies dominate the performance of LMMs. Modern multimodal models have demonstrated strong reasoning capabilities and the ability to provide contextually aligned answers, as evidenced by benchmarks like [1][2][3]. These capabilities suggest that those LMMs reason beyond simplistic biases.
> 2. **Intentional Benchmark Design**: Our benchmark is intentionally designed with direct, low-difficulty questions to disentangle and amplify the specific deficiencies—spurious inter-modality correlations and unimodal over-reliance. To further mitigate the influence of yes/no answering tendencies, we included a balanced distribution of yes/no ground-truth answers and diverse probing questions across objects and events. This ensures that the metrics reflect reasoning capabilities rather than systematic biases.
> 3. **Use of Yes/No Metrics in Prior Work**: Similar yes/no metrics have been widely adopted in prior work [4][5], such as POPE[4], which is widely adopted for evaluating hallucination tendencies in multimodal models, including proprietary systems like Gemini and GPT-4V, as well as open-source models like the LLaVA series. These benchmarks have proven effective in examining hallucination challenges while detecting tendencies toward yes/no biases. Furthermore, LMMs do not respond with simple *“yes”* or *“no”* but provide detailed answers, such as *“Yes, there is…”* or *“No, I didn’t hear…,”* reflecting reasoning rather than simplistic tendencies.
> 4. **Experimental Validation**: To further eliminate the potential influence of yes/no biases, we reformulated questions into multiple-choice formats (e.g., A. yes, B. no or A. no, B. yes) and randomized the mapping of answers across samples. This randomization ensures that models cannot simply rely on fixed answer tendencies. While most evaluated LMMs adhered to the reformulated instructions, some models (e.g., FAVOR and GroundingGPT) struggled to comply, opting instead to provide open-form answers directly addressing the question (e.g., *“Yes, there is…”* or *“No, I didn’t hear…”*). This behavior, consistent with observations in prior works [6][7], highlights limitations in instruction following for LMMs. This limitation reinforces the rationale for our intentional use of a straightforward format, designed to disentangle contributing factors and enable fair comparisons across baselines.
> Despite these challenges, the results show that the PA and HR metrics effectively capture the proposed deficiencies—spurious inter-modality correlations and unimodal over-reliance—rather than being confounded by yes/no tendencies.
> However, while this reformulation reduces yes/no bias, the original yes/no question format remains essential for detecting vulnerabilities specific to these tendencies. Such insights are critical for understanding and improving multimodal model behavior.
>
> In summary, our benchmark design ensures that the PA and HR metrics are robust and meaningful, capturing deficiencies like spurious inter-modality correlations and unimodal over-reliance. While revealing yes/no tendencies when present, our framework provides actionable insights into hallucination problems, aligning with the goals of systematic evaluation for multimodal models.
>
> | Model               | VL(pa/hr)   | AL(pa/hr)   | VAL(pa/hr)   | VDom(pa/hr)   | ADom(pa/hr)   | LDom(pa/hr)   | Overall(pa/hr)   |
> |:-                   |:-:          |:-:          |:-:           |:-:            |:-:            |:-:            |:-:               |
> | Gemini-1.5-flash    | 93.5/90.0   | 88.5/39.5   | 88.5/70.5    | 79.0/36.5     | 90.5/86.5     | 90.5/62.0     | 88.4/64.2        |
> | Gemini-1.5-flash (A/B) | 94.5/87.0 | 93.5/38.0   | 94.0/57.5    | 91.5/29.0     | 96.5/67.5     | 96.1/60.0     | 94.2/56.0          |
> | VideoLLaMA-2 (yes/no) | 75.0/86.0 | 77.5/94.0   | 78.0/98.0    | 62.0/75.5     | 80.0/90.0     | 57.5/43.0     | 71.7/81.1        |
> | VideoLLaMA-2 (A/B)       | 96.5/16.0   | 93.0/53.5   | 94.5/64.0    | 90.0/29.0     | 94.0/42.0     | 98.0/4.5      | 94.33/34.83      |

---

> ### Author Response · Authors · 2024-11-19
> **Reply to Weakness 2 (2/2)**
>
> [1] Yue, Xiang, et al. "Mmmu: A massive multi-discipline multimodal understanding and reasoning benchmark for expert agi." *CVPR*. 2024.
> [2] Lu, Pan et al. “MathVista: Evaluating Mathematical Reasoning of Foundation Models in Visual Contexts.” *ICLR* 2023.
> [3] Liu, Yuan, et al. "Mmbench: Is your multi-modal model an all-around player?." *ECCV*. Springer, Cham, 2025.
> [4] Li, Yifan, et al. "Evaluating Object Hallucination in Large Vision-Language Models." *EMNLP*. 2023.
> [5] Wang, Junyan et al. “Evaluation and Analysis of Hallucination in Large Vision-Language Models.” *ArXiv.* 2023.
> [6] Bitton, Yonatan, et al. "VisIT-Bench: a benchmark for vision-language instruction following inspired by real-world use." *NeurIPS*. 2023.
> [7] Pantazopoulos, Georgios, et al. "Learning To See But Forgetting To Follow: Visual Instruction Tuning Makes LLMs More Prone To Jailbreak Attacks." *LREC-COLING.* 2024.

---

> ### Author Response · Authors · 2024-11-19
> **Reply to Weakness 3**
>
> ### **Reply to Weakness 3: Relation to Rea-World Applications**
>
> We thank the reviewer for raising the concern about the relevance of our findings to real-world applications, particularly regarding the use of anti-common-sense videos for evaluating language dominance. Below, we provide clarifications:
>
> 1. **Relevance to Real-World Applications**: The findings from our benchmark directly address real-world challenges in multimodal applications such as embodied AI, robotics, and autonomous systems. These systems often encounter ambiguous or conflicting information across modalities, making it crucial to evaluate how well models can reason without over-relying on a dominant modality, such as language. The language dominance task uncovers vulnerabilities where LMMs prioritize text-based biases over visual/audio input, which can lead to critical failures in safety-critical applications.
> 2. **Rationale for Using Anti-Common-Sense Videos**: The use of anti-common-sense videos is inspired by established practices in LLM research, where “causal,” “common-sense,” and “anti-common-sense” samples are widely used to evaluate and mitigate hallucinations caused by language priors [1][2][3]. Such examples are designed to probe how models handle conflicts between their internal language-based expectations and external multimodal inputs. In our benchmark, anti-common-sense videos serve as straightforward and extreme cases to disentangle the causes of hallucinations and highlight specific vulnerabilities in language dominance. These examples are particularly effective for evaluating whether models overly rely on textual information when it contradicts evidence from other modalities, providing valuable insights for targeted improvements.
> 3. **Future Directions**: While this work focuses on controlled scenarios to systematically evaluate specific weaknesses, extending the benchmark to include more complex and nuanced real-world scenarios is an essential next step.
>
> The use of anti-common-sense videos in our benchmark is a deliberate and inspired design choice, aligning with established practices in LLM research. These examples effectively probe language dominance issues, systematically testing how well models resolve conflicts between language priors and multimodal evidence. This approach provides valuable insights into model vulnerabilities, ensuring that our findings remain relevant to improving multimodal systems in real-world applications. We will include this discussion in the main text to clarify the rationale behind this design choice.
>
> [1] Yao, Jia-Yu, et al. "LLM Lies: Hallucinations are not Bugs, but Features as Adversarial Examples.” *ArXiv*. 2023.
> [2] Jiang, Ling, et al. "Hallucination detection in LLM-enriched product listings." *LREC-COLING*. 2024.
> [3] Lei, Deren, et al. "Chain of natural language inference for reducing large language model ungrounded hallucinations." *ArXiv.* 2023.

---

> ### Author Response · Authors · 2024-11-19
> **Reply to Weakness 4**
>
> ### **Reply to Weakness 4: Consider Speech Setting**
>
> We appreciate the reviewer’s observation regarding the lack of speech-specific hallucination measurements and its potential impact on the scope of our work.
>
> 1. **Benchmark Design Philosophy:** Our benchmark is intentionally designed to be as simple and straightforward as possible, focusing on general daily-life scenarios where vision, audio, and language interplay. This approach ensures clear evaluation of multimodal reasoning in commonly encountered settings, aligning with our goal to disentangle and probe specific vulnerabilities in LMMs effectively.
> 2. **Speech in the Context of This Work:** Speech, as a specific sub-domain of audio, is also a distinct expression of language, making its modality definition somewhat ambiguous in our scope. Additionally, speech presents unique challenges, such as phoneme recognition, temporal synchronization with other modalities, prosody interpretation, and the complex interplay of semantics and acoustics. These challenges require dedicated datasets and evaluation frameworks tailored specifically to speech, which go beyond the general visual and audio event focus of this work. For these reasons, our benchmark does not explicitly address speech settings but prioritizes broader visual and audio events in general scenarios.
> 3. **Worth Exploring in Future Research:** While speech is not explicitly addressed in this work, it remains a valuable and distinct area worth exploring. The unique challenges and nuanced characteristics of speech settings require dedicated efforts, which we leave for further research in the field.
>
> In summary, our benchmark focuses on general daily-life multimodal scenarios, where speech is not explicitly addressed due to its unique challenges and overlapping categorization as audio or language. Nonetheless, it remains an important topic for future exploration in multimodal research.

---

> ### Author Response · Authors · 2024-11-19
> **Reply to Question 1**
>
> ### **Reply to Question 1: Applicability to Hallucination Detection**
>
> We appreciate the reviewer’s insightful question regarding the extension of our framework to detect partial hallucinations, such as those embedded within chain-of-thought reasoning. Below, we provide our response:
>
> 1. **Framework Purpose and Scope**: Our framework is designed to evaluate and categorize hallucinations systematically, focusing on fine-grained vulnerabilities such as spurious inter-modality correlations and unimodal over-reliance. Hallucination detection, including identifying partial hallucinations, is a distinct area of research (e.g., [1][2]) and lies outside the scope of this paper. Instead, our goal is to provide a robust evaluation framework that guides future research by highlighting critical deficiencies in LMMs.
> 2. **Applicability to Partial Hallucinations**: Although not explicitly designed for detection, our framework can still serve as a tool to analyze and understand hallucinations in random input-response pairs by assessing the influence of each category. For instance, in chain-of-thought reasoning, hallucinated segments can be examined to determine their underlying causes using our categorization.
> 3. **Future Directions**: Integrating detection methodologies with our evaluation framework offers a promising direction for future work. By extending our approach to incorporate partial hallucination detection, particularly in complex reasoning scenarios, we could provide a more comprehensive understanding of multimodal hallucinations and enhance the refinement of reasoning capabilities in LMMs.
>
> In summary, while our framework is focused on evaluation rather than detection, it can support the analysis and categorization of partial hallucinations, offering valuable insights to refine multimodal reasoning. Hallucination detection itself is a complementary area of research (e.g., [1][2]) that we believe can be integrated with our work to further advance this domain.
>
> [1] Chen, Xiang, et al. “Unified Hallucination Detection for Multimodal Large Language Models.” *ICLR Workshop*. 2024.
> [2] Su, Weihang, et al. “Unsupervised Real-Time Hallucination Detection based on the Internal States of Large Language Models.” *CoRR.* 2024.

---

> ### Author Response · Authors · 2024-11-19
> **Reply to Question 2**
>
> ### **Reply to Question 2: Incorporating Temporal Reasoning**
>
> We appreciate the reviewer’s insightful suggestion regarding the inclusion of temporal-related questions in our benchmark.
>
> 1. **Early Investigations**: During the early stages of this work, we explored temporal-related questions, such as absolute event timing and relative event sequences. However, these questions introduced additional complexities, significantly increasing task difficulty. Temporal reasoning requires LMMs not only to recognize events but also to analyze their sequential relationships and causal dynamics. Failures in these tasks often stemmed from general limitations in temporal reasoning capabilities rather than hallucination tendencies, making it challenging to isolate hallucination-specific causes.
> 2. **Benchmark Design Philosophy**: Our benchmark is intentionally designed to focus on straightforward, low-difficulty questions that emphasize object- and event-level existence. This ensures that wrong answers primarily reflect hallucination tendencies rather than broader reasoning limitations. Temporal-related questions involve higher cognitive demands, requiring robust event recognition and reasoning about sequential dynamics. As shown in existing temporal reasoning benchmarks[1][2][3], current LMMs often struggle with these tasks due to their design and insufficient temporal-related training data. Including such questions would complicate the evaluation process and deviate from our goal of disentangling specific hallucination causes.
>
> In summary, while temporal-related questions are valuable, they were excluded from our benchmark due to their complexity and the difficulty in isolating hallucination causes from general temporal reasoning failures. This decision aligns with our goal of providing a focused evaluation of hallucinations using straightforward tasks. Temporal reasoning remains an important area for dedicated benchmarks and future research.
>
> [1] Liu, Yuanxin, et al. "Tempcompass: Do video llms really understand videos?.” A*rXiv.* 2024.
> [2] Du, Yifan, et al. "Towards Event-oriented Long Video Understanding.” A*rXiv.* 2024.
> [3] Cai, Mu, et al. "TemporalBench: Benchmarking Fine-grained Temporal Understanding for Multimodal Video Models." A*rXiv.* 2024.

---

> ### Author Response · Authors · 2024-11-19
> **Reply to Question 3**
>
> ### **Reply to Question 3:** **Adapting the Benchmark for Improving LMMs**
>
> We thank the reviewer for their question regarding the adaptability of our benchmark in light of rapidly improving LMMs.
>
> 1. **Performance on VL Tasks and Focus on Tri-Modal Settings**: While VL LMMs demonstrate higher performance in certain subcategories, such as VL correlations, this reflects progress in the VL domain driven by specialized benchmarks and mitigation efforts. However, significant vulnerabilities persist, particularly in Language Dominance, where all open-source VL LMMs score below 70. This consistent underperformance highlights the challenges models face in overcoming inherent biases, such as language priors in LLM decoders. Moreover, these tasks in our benchmark serve as proof-of-concept validations to support our analysis of hallucination causes. For instance, language dominance evaluations highlight the persistent influence of language priors, where models overly rely on textual expectations over visual evidence. Such insights validate the robustness of our framework in identifying and categorizing hallucination types.
> However, our primary focus is on tri-modal (vision, audio, and language) settings, which present significantly more complex and realistic challenges. These scenarios better reflect real-world applications involving richer multimodal interactions. Although some VL models achieve strong results on VL subsets, training a tri-modal model that performs well across all subsets remains a non-trivial task. Our benchmark is designed to inspire the development of better tri-modal models that are both robust and practical for real-world use cases.
> 2. **Simplicity of Current Design and Persistent Challenges**: Our benchmark is intentionally designed with straightforward, low-difficulty tasks to isolate hallucination causes effectively. Despite this simplicity, even state-of-the-art proprietary models (e.g., Gemini 1.5) achieve average overall scores of only ~70%, highlighting persistent challenges in multimodal reasoning. This underscores the critical need for robust evaluation and targeted mitigation strategies to address these vulnerabilities, particularly in tri-modal settings.
> 3. **Continued Research and Future Updates**: While this work focuses on fundamental hallucination cases, future iterations of our benchmark could incorporate more complex scenarios, such as temporal dynamics and causal reasoning, as discussed in response to Question 2. These extensions would introduce new dimensions to multimodal evaluation, enabling deeper insights into hallucinations and broadening the benchmark’s applicability to real-world challenges.
>
> In summary, we respectfully disagree with the notion that our benchmark is no longer challenging for existing LMMs. While some VL tasks demonstrate high performance due to prior mitigation efforts in the VL domain, our tri-modal evaluations expose significant vulnerabilities that reflect complex, real-world scenarios. Our benchmark aims to inspire the development of more robust tri-modal models, and future iterations will continue to evolve to address emerging challenges in multimodal reasoning.

---

> ### Author Response · Authors · 2024-11-22
> **Follow-Up on Rebuttal Submission and Request for Feedback**
>
> Dear Reviewer 7vsK,
>
> We hope this message finds you well. Over two days ago, we submitted our detailed rebuttal addressing all your valuable feedback. We sincerely appreciate your insights and the opportunity to clarify and strengthen our work.
>
> We kindly request your attention to review our rebuttal and reconsider our work in light of the clarifications and evidence we provided. We are committed to ensuring that our rebuttal fully addresses your expectations.
>
> Thank you once again for your time and consideration.
>
> Best regards,
> The Authors

---

> ### Author Response · Authors · 2024-11-24
> **Request for Feedback Before Discussion Period Ends**
>
> Dear Reviewer 7vsK,
>
> As the author-reviewer discussion period approaches its end, we kindly request your feedback on our rebuttal. Your insights on whether we have effectively addressed your concerns would be greatly appreciated.
>
> We are truly grateful for the time and expertise you have dedicated to reviewing our work. Your thoughtful comments and suggestions have significantly contributed to improving the quality of our research.
>
> Thank you once again for your valuable feedback. We look forward to your guidance and hope to address any remaining concerns before the discussion period closes.
>
> Yours Sincerely,
> Authors

---

### Official Review · Reviewer_VdqN · 2024-11-04

**Soundness:** 2
**Presentation:** 3
**Contribution:** 3
**Rating:** 6
**Confidence:** 3

**Summary:**

This work provides a new way to understand and analyze hallucinations in multimodal models. Hallucinations are categorized into two main types with three subtypes under each main type.
A new benchmark is introduced base on this framework, providing 200 samples for each minor category. The benchmark is tested over several existing models, demonstrating the ability to distinguish the performance of these models. This might aid future researchers to analyze the hallucination in their own model.

**Strengths:**

This work provides an interesting attempt to categorize hallucinations in multimodal models by their causes. Most of the previous related work take hallucination as a single phenomenon and conducts unified assessments of them. This provides great help to researchers to locate problems in their model or in their training process, highlighting the value of this work as a benchmark. This is also a good try to encourage further discussion on hallucination in multimodal models.

**Weaknesses:**

1. Further experiment needed to support the categorization. The entire work is based on the assumption of a significant part of hallucinations in multimodal models is made up by the categories discussed in this work. Though there is experiment to prove that these categories exists and can explain some of the hallucinations, no statistical numbers is provided in this work to show how significant and identical they are.
2. Lack of Statistical numbers. This work doesn’t provide length distribution of its samples, neither a full distribution of the subjects/events presented in its sample.
3. 1200 sample/2400 questions is a big enough size for a test set, however 600 samples/1200 questions seems much smaller. Some models, like visual only or audio only models, can only use half of the dataset, making this work less helpful than it seems for some researchers.

**Questions:**

1. How can we categorize a hallucination? Not with your dataset, but with random input/response pair? Categorization in this work is a “principal content” one, reasons for different category is not independent. For example, the “hammer hitting” case provided in the work, is it a visual dominance hallucination as you categorize it, or it is actually a false visual-language correlation like “Chekhov's Gun”?
2. Human exists in every top 10 frequent subject pairs for visual cases. Any balancing methods are taken? Does it damage the soundness of your work?

---

> ### Author Response · Authors · 2024-11-19
> **Reply for Weakness 1 (1/2)**
>
> ### **Reply for Weakness 1: Significance and Distinction Support for Categorization**
>
> We appreciate the reviewer’s constructive feedback regarding the need for further experimental evidence to support our categorization of hallucinations. Below, we provide clarifications and additional insights:
>
> 1. **Significance of the Proposed Categories:** While we do not claim that our categorization encompasses all hallucination types, we present evidence that the proposed categories—inter-modality spurious correlations and unimodal overreliance—are both significant and prevalent. Prior studies in the vision-language domain have identified statistical training biases and language priors as primary causes of hallucinations [1][2][3][4], which served as the foundation for our work. We extend these findings to more complex tri-modality settings, extending statistical training biases to inter-modality spurious correlations and language priors to uni-modal over-reliance. Our categorization demonstrates their applicability and significance in multimodal contexts, with the potential for further extension to additional modalities.
> 2. **Basis for Categorization of sub-categories:**
>    - **Inter-Modality Spurious Correlations**: They arise from training biases inherent in multimodal datasets. Given that training data predominantly consists of visual-language (VL), audio-language (AL), and visual-audio-language (VAL) pairs, our categorization is a direct and logical reflection of this inherent structure, firmly rooted in the composition of multimodal datasets [5][6][7][8]. Figure 3 in our paper demonstrates a clear positive correlation between co-occurrence patterns in training data and hallucination rates, further supporting this categorization.
>    - **Uni-modal Over-reliance**: We beg to differ that our sub-categories under unimodal overreliance lack rigor. Given the three modalities (language, visual, audio), the sub-categories—language dominance, visual dominance, and audio dominance—are grounded in strong heuristics. These sub-categories represent distinct ways in which models fail by disproportionately relying on a single modality, a phenomenon systematically validated in our experiments. Moreover, we conducted additional experiments on the validation of Unimodal Overreliance similar to Figure 2. We sampled 20 failure cases with questions probing non-existent objects/events for each open-sourced LMM (where the LMM’s output probability  $p(\text{“yes”}|v, a, x) > p(\text{“no”}|v, a, x)$ on those test cases , but the ground truth answer is *“no”*). We plotted the average  $p(\text{“yes”}/\text{“no”}|v, a, x)$  along with standard deviations across blur/noise steps applied to specific input modalities. The results, plotted with average probabilities and standard deviations across blur/noise steps, align with our original findings and highlight the distinct challenges posed by specific modality overreliance, further supporting our categorization.
> (Links to reulst figures: https://ibb.co/dWnVF6R , https://ibb.co/Z6qYFnd , https://ibb.co/YQsNbR7)
>
> 3. **Benchmark Design and Experimental Insights**: Our benchmark is explicitly designed to isolate and amplify the impact of specific sub-categories by employing clear and targeted question formats. By focusing on object- and event-level existence probing, it removes extraneous complexities present in other multimodal benchmarks, ensuring that observed hallucinations can be directly attributed to the targeted categories. For example, V-L correlation samples are constructed using top co-occurrence patterns from datasets like WebVid10M to emphasize inter-modality spurious correlations. Results in Figure 3 and Tables 2, 4, and 5 demonstrate that even state-of-the-art proprietary models exhibit severe hallucinations in these straightforward cases, validating our categorization and underscoring the importance of addressing these challenges.
>
> In summary, while additional hallucination types may exist, our work demonstrates that inter-modality spurious correlations and unimodal overreliance are critical and under-explored phenomena. Our benchmark design, experimental findings, and categorization offer a strong foundation for future advancements in understanding and mitigating hallucinations in multimodal systems. We appreciate the reviewer’s feedback, which provides valuable guidance for refining our approach and further validating our contributions.

---

> ### Author Response · Authors · 2024-11-19
> **Reply for Weakness 1 (2/2)**
>
> [1] Zhou, Yiyang, et al. "Analyzing and Mitigating Object Hallucination in Large Vision-Language Models." *ICLR*. 2024.
> [2] Leng, Sicong, et al. "Mitigating object hallucinations in large vision-language models through visual contrastive decoding." *CVPR*. 2024.
> [3] Li, Yifan, et al. "Evaluating Object Hallucination in Large Vision-Language Models." *EMNLP*. 2023.
> [4] Chang, Yue, et al. "A Unified Hallucination Mitigation Framework for Large Vision-Language Models." *TMLR*. 2024.
> [5] Wang, Yi, et al. "InternVid: A Large-scale Video-Text Dataset for Multimodal Understanding and Generation." *ICLR*. 2024.
> [6] Bain, Max, et al. "Frozen in time: A joint video and image encoder for end-to-end retrieval." *ICCV*. 2021.
> [7] Sun, Luoyi, et al. "Auto-ACD: A large-scale dataset for audio-language representation learning." *ACMMM*. 2024.
> [8] Kim, Chris Dongjoo, et al. "Audiocaps: Generating captions for audios in the wild." *NAACL*. 2019.

---

> ### Author Response · Authors · 2024-11-19
> **Reply for Weakness 2**
>
> ### **Reply for Weakness 2: Lack of Benchmark Statistics**
>
> We appreciate the reviewer’s suggestion to enhance the statistical analysis of our dataset. To address this, we will include the following figures and details in the appendix to provide a more comprehensive understanding of the sample distributions in our work:
>
> 1. **Audio and Video Length Distribution**: We provide three figures illustrating the length distributions for each sample type in our dataset:
>
>     - **Video-only**: This subset contains 800 samples with video content only. (Link to figure: https://ibb.co/k5gYDbT)
>
>     - **Audio-only**: This subset contains 400 samples with audio content only. To clarify, as stated in our paper, the AL spurious correlations and corresponding test samples are extracted and sampled from the large-scale Audio-Language training dataset Auto-ACD. All samples in Auto-ACD contain approximately 10 seconds of audio, and the length distribution of our Audio-Only samples (specific to the AL Spurious Correlations subcategory) follows this characteristic. (Link to figure: https://ibb.co/xq6DdCH)
>
>     - **Video-Audio Pairs**: This subset comprises 1200 samples with synchronized audio and video. (Link to figure: https://ibb.co/4McXfb7)
>
> 2. **Object and Event Distribution**: We have generated three figures to show the full distributions of object frequencies, visual event frequencies, and audio event frequencies.
>
>     - To clarify, while the entire benchmark does not aim to strictly replicate real-world distribution, the co-occurrence and frequent appearance patterns used to construct the benchmark are derived from the large-scale dataset WebVid10M, which reflects real-world data distributions. These patterns are selected to emphasize specific correlations, ensuring the benchmark rigorously probes LMMs with fine-grained categorizations.
>
>     - Although sampling from frequent patterns may modify the overall distribution of the benchmark, our intent is not to replicate the entire natural long-tail distribution but to isolate and amplify high-impact cases. This approach maintains the authenticity of the correlations being probed while ensuring the benchmark effectively evaluates spurious inter-modality correlations in realistic contexts.
>    (Links to figures: https://ibb.co/Ttt2q9D, https://ibb.co/F6wzZkC , https://ibb.co/bbf9fM2)
>
> 3. **Statistical Figures in Appendix:** A more fine-grained distribution, showcasing the top 10 objects/events separated by existent/non-existent probing, is also originally provided in Appendix Figure 4.
>
> To summarize, in response to the reviewer’s suggestion, we have added detailed figures and analyses to provide a comprehensive view of our dataset’s distributions. These include length distributions for different audio and video types, as well as full and fine-grained distributions of object and event frequencies. We have also clarified how the long-tail distribution, derived from real-world patterns, supports the reliability of our findings. This enhances the statistical robustness of our dataset and strengthens the benchmark’s alignment with real-world scenarios.

---

> ### Author Response · Authors · 2024-11-19
> **Reply for Weakness 3**
>
> ### **Reply for Weakness 3: Concerns on Benchmark Subset Sizes**
>
> We appreciate the reviewer’s feedback regarding the test set size for VL and AL LMMs. We would like to address this concern as follows:
>
> 1. **Primary Focus on VAL LMMs**: Our benchmark is designed to evaluate the reasoning and integration capabilities of LMMs across language, visual, and audio modalities, rather than serve as a training dataset. VAL LMMs are crucial as they integrate the three most common modalities of human perception—vision, audio, and language—which are essential for comprehending and interacting with the real world. This tri-modal integration is fundamental to real-world applications such as embodied AI, robotics, and autonomous driving, where understanding and reasoning across modalities are necessary for robust decision-making and situational awareness.
>
> 2. **Proof-of-Concept on VL and AL LMMs**: While several hallucination benchmarks exist for VL-only and AL-only LMMs [1][2][3][4], our evaluation of those LMMs serves as a proof-of-concept to validate our analysis of hallucination causes and the effectiveness of our categorization framework.
> For instance, our evaluation of spurious correlations in VL and AL models confirms that such correlations primarily stem from training data biases. The results also validate our categorization of spurious correlations into three subcategories—VL, AL, and VAL—each addressing unique inter-modality challenges. Specifically, VL models often exhibit hallucinations due to visual-text co-occurrence biases, whereas AL models are influenced by biases in audio-text mappings. These findings underscore the robustness and utility of our categorization framework for systematically analyzing hallucination causes across modalities.
>
> 3. **Scalability and Future Expansion**: Our benchmark data construction process is inherently scalable. In the final version, we will scale up the test sets for VL-only and AL-only models to match the size of the VAL test set, providing 1,200 samples and 2,400 probing questions for each setting. This expansion will enhance the benchmark’s utility, enabling comprehensive evaluations for VL, AL, and VAL LMMs, and making it more helpful for a broader range of research scenarios and LMMs.
>
> In summary, we thank the reviewer for the valuable suggestion regarding the size of the test set for single-modality models. While our primary focus is on VAL LMMs, given their critical role in real-world applications, we also recognize the importance of supporting VL and AL LMMs. Our evaluations of VL and AL models also serve as proof of concept, validating our analysis of hallucination causes and the robustness of our categorization framework. To address the reviewer’s concern, we will scale up the test sets for VL and AL models to match the VAL test set size in the final version, ensuring broader applicability and utility for diverse research scenarios and LMMs. We believe these improvements will significantly enhance the benchmark’s value to the research community.
>
> [1] Li, Yifan, et al. "Evaluating Object Hallucination in Large Vision-Language Models." *EMNLP*. 2023.
> [2] Ding, Peng, et al. "Hallu-pi: Evaluating hallucination in multi-modal large language models within perturbed inputs." *ACMMM*. 2024.
> [3] Kuan, Chun-Yi, and Hung-yi Lee. "Can Large Audio-Language Models Truly Hear? Tackling Hallucinations with Multi-Task Assessment and Stepwise Audio Reasoning." *ArXiv*. 2024.
> [4] Nishimura, Taichi, Shota Nakada, and Masayoshi Kondo. "On the audio hallucinations in large audio-video language models." *ArXiv*. 2024.

---

> ### Author Response · Authors · 2024-11-19
> **Reply to Question 1**
>
> ### **Reply to Question 1: Hallucination Categorization of Random Cases**
>
> We appreciate the reviewer’s thoughtful question regarding hallucination categorization. Below, we address the complexity and scope of our approach:
>
> 1. **Categorization Framework and Purpose**: We acknowledge that hallucinations in real-world scenarios often stem from interrelated factors. Our categorization—grounded in prior works and validated through experiments—is not designed to comprehensively detect or classify hallucinations in all scenarios. Instead, it provides a systematic framework for evaluating large multimodal models (LMMs) and identifying fine-grained vulnerabilities. Importantly, hallucination detection itself is a distinct area of research (e.g., [1][2]), which involves developing algorithms to automatically detect hallucinations in random input-response pairs. Our focus is on evaluation and analysis, leaving detection as a complementary but separate line of work.
> 2. **Benchmark Design and Detection Applicability**: As detailed in response to Weakness 1, our benchmark is intentionally designed to minimize ambiguities by employing direct and simplified probing questions with low task difficulty. This design isolates and amplifies the impact of specific sub-categories, ensuring that each sample primarily reflects a single dominant hallucination cause. While the categorization framework is tailored for the controlled environment of our benchmark, it can still serve as a tool to analyze and understand hallucinations in random input-response pairs by assessing the influence of each category.
> 3. **Chekhov’s Gun Example**: The Chekhov’s Gun example is an interesting case that illustrates how multiple factors may interact to cause hallucinations. If a model hallucinates a visual event like *“gun firing”* from *“a gun placed on a table,”* this could partially result from event-level visual-language correlations in the training data that associate objects (e.g., a gun) with specific events (e.g., firing). However, the hallucination of an audio event like *“hearing gunfire”* may involve an additional layer of visual dominance, where the model overrelies on visual inputs and ignores the lack of supporting audio evidence. This type of hallucination may not be fully attributed to visual-language correlations alone, as VL training data often lacks textual descriptions for auditory events, but only visual ones. While our analysis here is not strictly grounded, it highlights the nuanced interactions between different modalities and underscores the need for further research to disentangle such complex phenomena.
>
> In summary, we thank the reviewer for raising this important question. While real-world hallucinations often involve multiple interrelated factors, our categorization framework provides an effective and systematic means to analyze and evaluate LMMs. By simplifying our benchmark design to isolate specific sub-categories, we ensure clear insights into the primary causes of hallucinations. Furthermore, hallucination detection itself is a distinct area of research (e.g., [1][2]), which complements our work and can further enhance the understanding and mitigation of hallucinations in multimodal systems.
>
> [1] Chen, Xiang, et al. "Unified Hallucination Detection for Multimodal Large Language Models." *ICLR Workshop*. 2024.
> [2] Su, Weihang, et al. "Unsupervised Real-Time Hallucination Detection based on the Internal States of Large Language Models." *CoRR.* 2024.

---

> ### Author Response · Authors · 2024-11-19
> **Reply to Question 2**
>
> ### **Reply to Question 2: “Human” Dominance in VL Correlations**
>
> We appreciate the reviewer’s question regarding the dominance of “human” in VL correlations and its potential impact on the balance and soundness of our benchmark. We address this concern as follows:
>
> 1. **Spurious Correlation Distribution:** The dominance of “human” in VL event-level correlations reflects real-world data distributions, as humans are central to most visual scenarios. These correlations are formed from over-represented patterns in training data, such as those extracted from WebVid10M, a large-scale video-text dataset. This inherent bias is not only natural but also intentional in our benchmark design. By including these over-represented patterns, we aim to better disentangle the factors contributing to hallucinations and evaluate how models handle biases and spurious correlations that are prevalent in real-world multimodal datasets.
> 2. **Balancing Strategies**:
>     - For categories where spurious correlations are dominated by “human” (e.g., VL object-level co-occurrences and event-level appearances and co-occurrences), the dataset reflects this real-world bias to evaluate how models handle such scenarios.
>     - For other categories, such as AL and VAL correlations and unimodal overreliance, we manually ensured the inclusion of examples covering various categories beyond human-centric pairs (e.g., beach, car, flower, dog, city, etc). This approach diversifies the test set and broadens the evaluation scope, capturing different real-world contexts.
> 3. **Impact on Soundness**: The prevalence of “human” does not compromise the soundness of our work. Instead, it enhances the benchmark’s ability to reflect real-world distributions and evaluate the impact of over-represented patterns on model performance. By combining the intentional inclusion of such patterns with balanced and diverse examples in other categories, we ensure that the benchmark remains comprehensive, reliable, and effective for evaluating hallucinations in LMMs across a variety of contexts.
>
> We thank the reviewer for raising this important point and believe that our approach ensures robustness while addressing spurious correlations and biases in real-world multimodal datasets.

---

> ### Author Response · Authors · 2024-11-22
> **Follow-Up on Rebuttal Submission and Request for Feedback**
>
> Dear Reviewer VdqN,
>
> We hope this message finds you well. Over two days ago, we submitted our detailed rebuttal addressing all your valuable feedback. We sincerely appreciate your insights and the opportunity to clarify and strengthen our work.
>
> We kindly request your attention to review our rebuttal and reconsider our work in light of the clarifications and evidence we provided. We are committed to ensuring that our rebuttal fully addresses your expectations.
>
> Thank you once again for your time and consideration.
>
> Best regards,
> The Authors

---

> ### Author Response · Authors · 2024-11-24
> **Request for Feedback Before Discussion Period Ends**
>
> Dear Reviewer VdqN,
>
> As the author-reviewer discussion period approaches its end, we kindly request your feedback on our rebuttal. Your insights on whether we have effectively addressed your concerns would be greatly appreciated.
>
> We are truly grateful for the time and expertise you have dedicated to reviewing our work. Your thoughtful comments and suggestions have significantly contributed to improving the quality of our research.
>
> Thank you once again for your valuable feedback. We look forward to your guidance and hope to address any remaining concerns before the discussion period closes.
>
> Yours Sincerely,
> Authors

---

> > ### Comment · Reviewer_VdqN · 2024-11-25
> > **Response to the rebuttal from the authors**
> >
> > Dear Authors,
> >
> > Thank you for your message and your thoughtful rebuttal. I appreciate the effort you have put into addressing my comments and concerns. After carefully reviewing your responses, I acknowledge the points raised and the improvements made to clarify your work.
> >
> > However, after further consideration, I believe my initial evaluation and rating remain consistent with my overall assessment of the paper. While your rebuttal provides helpful context, it does not fully address the key aspects of my critique.
> >
> > Thank you once again for your efforts, and I wish you the best with the continued development of your research.
> >
> > Best regards,
> > Reviewer VdqN

---

> > > ### Author Response · Authors · 2024-11-25
> > > **Follow-Up on Reviewer Feedback**
> > >
> > > Dear Reviewer VdqN,
> > >
> > > Thank you for taking the time to review our rebuttal. We sincerely appreciate your acknowledgment of the improvements we made and your careful consideration of our responses.
> > >
> > > We understand that certain aspects of your critique remain unresolved. Could you kindly specify the key points you feel were insufficiently addressed and their importance? This clarity would greatly help us better address your concerns and improve our work.
> > >
> > > We deeply value the author-reviewer discussion period as an opportunity for constructive dialogue, and we sincerely appreciate your time and feedback.
> > >
> > > Best regards,
> > > Authors

---

> > > ### Author Response · Authors · 2024-11-29
> > > **Follow-Up on Unresolved Concerns**
> > >
> > > Dear Reviewer VdqN,
> > >
> > > Thank you once again for your thoughtful feedback and for carefully reviewing our rebuttal. We appreciate your acknowledgment of the improvements made and understand that certain aspects of your critique remain unresolved.
> > >
> > > If possible, could you provide additional clarification or specify which key aspects of your concerns were not fully addressed? Understanding these points in greater detail would be invaluable for us to refine our approach and strengthen our work further.
> > >
> > > We deeply respect the insights you have shared and remain committed to addressing your concerns to the best of our ability. We greatly appreciate the positive feedback you have provided, which has encouraged us in refining and presenting our work. Your guidance has been instrumental in shaping the quality of this submission, and we sincerely hope to benefit from your continued input.
> > >
> > > Thank you again for your time and consideration.
> > >
> > > Best regards,
> > > The Authors

---

### Official Review · Reviewer_5Kmo · 2024-11-04

**Soundness:** 3
**Presentation:** 3
**Contribution:** 3
**Rating:** 6
**Confidence:** 4

**Summary:**

This paper focuses on the phenomenon of hallucinations in multimodal large language models (MLLMs), establishing benchmarks and dataset, while also suggesting future research directions. Overall, the paper is well-written and clearly presented.

**Strengths:**

Hallucination is an important research topic in LLMs. This study expands previous research on hallucinations from purely linguistic tasks to multi-modal tasks, categorizing the causes into two aspects: excessive reliance on unimodal priors and false inter-modal correlations.

**Weaknesses:**

However, I still have some concerns regarding this work:

1.	My primary focus is on the benchmark dataset. Multimodal data may encounter issues of modality mutual exclusion. For example, audio may express one content while video expresses another content. Thus, when we combine multimodal information, it may be difficult to obtain a singular ground truth, which may impact the reliability of this work.

2. In this paper, the model takes all available modalities as inputs. However, if the prompts focus solely on auditory clues, can we directly input audio into the model to simply address the hallucination issue? For example, in Figure 1(b), since the prompt focuses on auditory cues, can we resolve the hallucination issue by only inputting audio data? Therefore, I wonder if there is a simpler approach to tackle the hallucination problem, which would also avoid the first issue of modality mutual exclusion.

**Questions:**

Please refer to my comments on weaknesses.

---

> ### Author Response · Authors · 2024-11-19
> **Reply to Weakness 1**
>
> ### **Reply for Weakness 1: Issue of Modality Mutual Exclusion**
>
> Thank you for your insightful feedback regarding potential modality mutual exclusion in our dataset. We would address your concerns from two aspects:
>
> 1. **Ground Truth Consistency**: To ensure a robust and reliable benchmark, we have manually reviewed each data sample in our benchmark to confirm that the audio, visual, and language modalities are contextually aligned. This review process verifies alignment and ensures that there is only one definitive ground truth for each of our probing questions.
> 2. **Modality Mutual Exclusion & Hallucination**: In our study, we define hallucination as the discrepancy between the factual multimodal input and the generated textual output. While the indicated modality mutual exclusion—where audio and video convey different content—is an interesting area for future exploration, we consider it orthogonal to hallucination. Specifically, while audio and video express distinct content, our expectation for reliable LMMs is that their generated audio-related textual content should accurately align with the audio input, and video-related textual content should align with the video input. Audio-aligned content that diverges from the video, or vice versa, would not be categorized as a hallucination under the current scope.
>
> We appreciate the suggestion and recognize the potential for future work exploring this nuanced interaction in multimodal datasets.

---

> ### Author Response · Authors · 2024-11-19
> **Reply for Weakness 2**
>
> ### **Reply for Weakness 2: "Simple" Mitigation Strategy**
>
> Thank you for highlighting the possibility of addressing hallucination by selectively inputting specific modalities based on the prompt focus. We would share our views on this issue from three aspects:
>
> 1. **Synergistic Benefits of Full Multimodal Input**: Exploiting complementary multimodal information is critical for more reliable and accurate MLLM generation. In scenarios where one modality may be limited or ambiguous, other modalities can provide valuable supplementary cues. For instance, if a video is low-resolution or contains objects that are hard to recognize visually (e.g., a small fuzzy animal back-facing the camera), audio cues like barking or meowing can help resolve ambiguity and ensure correct recognition. By leveraging all available modalities, models can achieve a more holistic understanding and generate responses that better reflect the input context. This synergistic effect highlights the importance of full multimodal input, even in cases where a single modality may seem sufficient for the prompt focus.
>
> 2. **Challenges in Selective Modality Use**: While focusing on a single modality may reduce hallucination in some scenarios, selectively discarding modalities during inference introduces significant challenges. Existing multimodal systems are not designed to determine, at runtime, which modality holds the most relevant information or to preemptively ignore others. Multimodal inputs often involve complex interactions, such as modality dominance or mutual exclusion, where information relevance may shift unpredictably. For example, even if the prompt targets auditory cues, visual or textual inputs may still provide crucial context or disambiguation. Ensuring cohesive integration across all modalities remains essential for robust and adaptable real-world applications.
>
> 3. **Experimental Validation**: In response to the reviewer’s suggestion, we conducted two additional experiments:
>    - **LMM Behavior Analysis**: Similar to Figure 2, we sampled 20 failure cases with questions probing non-existent objects/events for each open-sourced LMM under the Visual and Audio Dominance subcategories (where the LMM’s output probability  $p(\text{“yes”}|v, a, x) > p(\text{“no”}|v, a, x)$ on those test cases , but the ground truth answer is *“no”*). We plotted the average  $p(\text{“yes”}/\text{“no”}|v, a, x)$  along with standard deviations across blur/noise steps applied to specific input modalities. The observed trends align with our original findings in Figure 2, demonstrating that reducing information from the dominant modality forces the model to rely more on the targeted modality, effectively decreasing hallucination rates. This highlights the critical impact of modality dominance on hallucinations and underscores the necessity of robust cross-modal integration, particularly in scenarios involving mutual exclusion.(Links to results figures: https://ibb.co/Z6qYFnd , https://ibb.co/YQsNbR7)
>    - **Quantitative Evaluation**: Furthermore, instead of discarding non-focused modalities, we prompted LMMs to prioritize specific modalities based on the question (e.g., *“Please focus more on the given audio information to answer the question.”* for *“Did you hear…”* queries) while retaining inputs from all modalities to preserve complementary information. Experimental results show that while this approach reduced hallucinations to some extent, the improvements were limited and highly context-dependent, reinforcing the challenges of achieving robust multimodal reasoning through selective prompting.
>
>    |Model|Visual Dom(pa/hr)|Audio Dom(pa/hr)|
>    |:-|:-:|:-:|
>    |Gemini-1.5-flash|79.0/36.5|90.5/86.5|
>    |+focus prompt|94.5/29.0|95.0/68.5|
>    |GroundingGPT|99.5/1.0|98.5/23.5|
>    |+focus prompt|96.0/1.5|96.5/41.5|
>    |FAVOR|89.0/21.5|92.0/43.5|
>    |+focus prompt|88.0/30.0|89.5/57.5|
>    |VideoLLaMA2|62.0/75.5|92.0/43.5|
>    |+focus prompt|67.0/52.5|83.0/81.5|
>
> In summary, while selectively focusing on specific modalities may appear promising, the synergistic benefits of full multimodal input and the challenges of dynamically determining modality relevance underscore the necessity of cohesive integration. Our experiments validate the critical role of cross-modal reasoning and reveal the limitations of selective or prompted approaches. The tri-modal setting in our benchmark provides a comprehensive evaluation of multimodal models, highlighting their limitations and emphasizing the need for more robust and adaptable reasoning capabilities. We appreciate your suggestion, which provides valuable insights for refining our methodology and guiding future research directions.

---

> ### Author Response · Authors · 2024-11-22
> **Follow-Up on Rebuttal Submission and Request for Feedback**
>
> Dear Reviewer 5Kmo,
>
> We hope this message finds you well. Over two days ago, we submitted our detailed rebuttal addressing all your valuable feedback. We sincerely appreciate your insights and the opportunity to clarify and strengthen our work.
>
> We kindly request your attention to review our rebuttal and reconsider our work in light of the clarifications and evidence we provided. We are committed to ensuring that our rebuttal fully addresses your expectations.
>
> Thank you once again for your time and consideration.
>
> Best regards,
> The Authors

---

> ### Author Response · Authors · 2024-11-24
> **Request for Feedback Before Discussion Period Ends**
>
> Dear Reviewer 5Kmo,
>
> As the author-reviewer discussion period approaches its end, we kindly request your feedback on our rebuttal. Your insights on whether we have effectively addressed your concerns would be greatly appreciated.
>
> We are truly grateful for the time and expertise you have dedicated to reviewing our work. Your thoughtful comments and suggestions have significantly contributed to improving the quality of our research.
>
> Thank you once again for your valuable feedback. We look forward to your guidance and hope to address any remaining concerns before the discussion period closes.
>
> Yours Sincerely,
> Authors

---

> ### Author Response · Authors · 2024-11-28
> **Thank You and Request for Further Feedback**
>
> Dear Reviewer 5Kmo,
>
> Thank you for your positive review and for recognizing the relevance and novelty of our benchmark. We greatly appreciate the time and effort you have dedicated to improving our work.
>
> We hope our detailed rebuttal has effectively addressed your concerns. If there are any remaining questions or additional clarifications you would like us to provide, we would be happy to assist.
>
> Your insights have been invaluable, and we truly appreciate your guidance throughout this process.
>
> Best regards,
> The Authors

---

### Author Response · Authors · 2024-12-03
**Summary of Reviewer Feedback and Rebuttal Response**

Dear Reviewers and Area Chair,

We sincerely thank all reviewers for their thoughtful feedback and valuable insights, which have greatly enhanced the quality and clarity of our work.

We are encouraged by the recognition of our contributions, including:
- Introducing a novel and relevant benchmark for evaluating hallucinations in multimodal large language models (LMMs), spanning vision, audio, and language modalities.
- Providing a clear and systematic categorization of hallucination causes, including spurious inter-modality correlations and unimodal over-reliance.
- Offering actionable insights and establishing a foundation for future advancements in multimodal hallucination research.

In the rebuttal, we thoroughly addressed key concerns raised during the review process, including:
- Dataset Statistics and Limitations: We included detailed figures and explanations, along with discussions on scale, diversity, and potential biases in our benchmark.
- Experimental Validation: We provided additional experiments and evidence to substantiate our claims about modality dominance and spurious correlations.
- Adaptability to Advancing LMMs: We clarified the benchmark’s relevance and challenges for tri-modal models, emphasizing the future potential to accommodate more complex scenarios.

These efforts were acknowledged by most reviewers, who noted that their concerns were well addressed and expressed appreciation for our responses.

For `Reviewer XxQu`, we are grateful for their recognition of our contributions and acknowledgment of the improvements made. While no further rating adjustment was provided during the discussion period and there was no further reply from the reviewer after we addressed their concerns, we have carefully and comprehensively responded to all points raised in their review and rebuttal comments.

We deeply value the opportunity to engage in this constructive dialogue and appreciate the time and effort all reviewers have dedicated to this process. Your guidance and feedback have been instrumental in shaping our work, and we remain committed to building on these insights in future research.

Thank you once again for your support and consideration.

Best regards,
The Authors

---

### Meta-Review · Area_Chair_6xc2 · 2024-12-20

**Metareview:**

This work provides a new way to understand and analyze hallucinations in multimodal models. A new benchmark is introduced based on this framework, and is used to evaluate several existing multimodal models.

Reviewers generally found the task relevant and novel, and that it will be useful for evaluating multimodal models, and the results and analysis provided in the paper.

However, there were also shared concerns regarding dataset size, scale, and diversity, the strong performance of many baselines, and the lack of addressing key multimodal challenges with the datasets. I have carefully reviewed the discussions and feel that many key points remain unaddressed, including the possible lack of established and rigorously annotated ground truth in the dataset, the relative small size, scale, and diversity (I know the authors argued that it's larger than some of the current ones, but it still seems insufficient to justify reliable research). Finally, another critical issue is the lack of analysis on what makes this benchmark truly 'multimodal' beyond an ensemble of unimodal hallucination benchmarks. There is no analysis of how the modalities interact with each other - do they provide similar, independent, or contradicting information, as raised by reviewer 5Kmo. In light of these concerns, I recommend rejection.

**Additional Comments On Reviewer Discussion:**

Reviewer 7vsK raised their score from 5 to 6 after the discussion since they felt their main concerns had been addressed, which I agree. Reviewers VdqN and 5Kmo had a score of 6 and did not change it during the discussion. Reviewer XxQu maintained a score of 5 after the discussion. They primarily pointed out issues with dataset size, scale and diversity of the data, already strong performance of many baselines, and the lack of addressing the multimodal challenges with the datasets. I have carefully reviewed the discussions and feel that many key points remain unaddressed:

1. For example, reviewer 5Kmo pointed out that 'when we combine multimodal information, it may be difficult to obtain a singular ground truth, which may impact the reliability of this work', which is a very relevant point. The authors respond with 'Ground Truth Consistency: To ensure a robust and reliable benchmark, we have manually reviewed each data sample in our benchmark to confirm that the audio, visual, and language modalities are contextually aligned.' This is not a satisfactory answer - there were no concrete explanations of how the 'manual review' was done. I'm guessing this was done by the authors, which is not ok since there is a conflict of interest, instead it should be manually reviewed by external annotators who are instructed carefully and paid a fair amount. Actually doing this experiment and ensuring the reliability of the ground truth is critical.

2. In response to reviewer 5Kmo, the authors also point out that 'where audio and video convey different content—is an interesting area for future exploration, we consider it orthogonal to hallucination'. I again disagree - these are exactly cases where the multimodal contribution is interesting, otherwise the benchmark just turns into an ensemble of unimodal hallucination benchmarks. This is an issue throughout the discussion, where the 'multimodal' nature of the benchmark has not been given enough appreciation.

3. For reviewer VdqN and XxQu's concerns on dataset size, scale, and diversity, I also feel that these points have not been sufficiently addressed. I know the authors argued that it's larger than some of the current ones, but it still seems insufficient to justify reliable research. The authors can attempt to scale up the data collection to make this a truly comprehensive and large-scale benchmark.

---

### Decision · Program_Chairs · 2025-01-22

Reject